# Bridging Graph Worlds: Neural Approximation of Gromov-Wasserstein Distances

## Abstract

Graph-structured data is crucial in various domains like biology and networks. Comparing graphs, which is a fundamental problem in graph data analysis, is nonetheless highly challenging. Recently, the Gromov-Wasserstein (GW) distance has provided a principled way to compare two graphs. However, computing the GW distance involves solving a complex non-convex optimization problem, making it computationally expensive, especially when the graphs are large. In this work, we propose a neural approximation of the GW distance, called NeuralGW. In NeuralGW, we use a combination of a graph isomorphism network and a transformer to represent the nodes of two graphs as two sets of vectors, treated as two discrete distributions, on which we compute multiple maximum mean discrepancy values given by different kernels. We then use a multilayer perceptron to convert the vector formed by these values into a single value, which is the prediction of the GW distance. Once trained, the model allows for efficient inference, enabling fast structural comparisons between graphs across diverse domains. We also provide a theoretical guarantee for the generalization ability of NeuralGW. Experiments demonstrate the effectiveness and practical applicability of our approach on real-world datasets, in comparison to baselines.

## 1 Introduction

Graph-structured data arises naturally in a wide range of domains like chemistry (Rong et al., 2020), biology (Agarwal, 2006), social networks (Aggarwal, 2011), knowledge graphs (Hogan et al., 2021), and computer vision (Riesen & Bunke, 2008). In the general sense, a graph represents some entities and their relationships. Comparing such graph objects—particularly when structural similarity is more meaningful than node-level features—is a core problem in machine learning, such as classification, clustering, and graph retrieval. However, graph comparison is inherently challenging due to the lack of canonical node orderings, varying sizes, and the potential absence of node correspondences. Accurate and efficient graph comparison thus remains a crucial open problem.

Several common approaches exist for comparing pairs of graphs. Classical non-parametric graph kernels, such as the random walk kernel (Vishwanathan et al., 2010), the Weisfeiler-Lehman kernel (Shervashidze et al., 2011), and the Weisfeiler-Lehman optimal assignment kernel (Kriege et al., 2016), represent widely used techniques. Alternatively, optimal transport (OT)-based methods frame a graph as an empirical distribution (Cuturi, 2013). The Wasserstein distance, for instance, offers a principled framework for comparing distributions by accounting for the underlying geometry of the sample space (Villani et al., 2009). Despite their theoretical appeal, OT-based methods and many graph kernels are often computationally expensive in practice. To address these scalability issues, deep learning-based approaches have been developed. Courty et al. (2017), for example, introduced a CNN-based Wasserstein embedding that approximates the OT distance with a more efficient Euclidean distance. More recently, Haviv et al. (2024) designed a Transformer-based "Wasserstein Wormhole" architecture for the same purpose. A significant limitation of these neural OT approximators, however, is their requirement for a shared input domain; they cannot directly compare graph pairs with heterogeneous node feature dimensions. It is also worth noting that unsupervised graph representation learning methods (Sun et al., 2020; You et al., 2020; Sun et al., 2023) enable indirect graph comparison. These techniques allow for the computation of a Euclidean distance between the learned representation vectors of different graphs, providing another viable comparison strategy.

The Gromov-Wasserstein (GW) distance (Mémoli, 2011) extends the classical OT framework (Villani et al., 2009) by seeking a coupling that aligns the pairwise relational structures of two spaces. In contrast to the network GW of (Chowdhury & Mémoli, 2019), the adjacency matrix of an unweighted and undirected graph naturally induce a metric structure, allowing such graphs to be treated directly within the standard GW framework on measurable metric spaces. This viewpoint offers a general and coherent foundation for comparing graphs through their intrinsic geometries. Since GW operates on pairwise distances rather than their absolute coordinates, it captures structural similarity in a way that is particularly well suited for graph comparison (Xu et al., 2019; 2022). In this sense, GW distance provides a principled bridge for comparing graphs.

Despite its theoretical appeal, GW distance also suffers from its high computational complexity $\mathcal{O}(n^4)$ or $\mathcal{O}(n^3)$ (Peyré et al., 2016) since it is a non-convex quadratic minimization problem. Xu et al. (2019) proposed to use proximal point algorithm (PPA) for solving the GW distance while Li et al. (2023a) introduced Bregman Alternated Projected Gradient (BAPG) method for GW computation. To improve its scalability, Scetbon et al. (2022) shows that imposing low-rank assumptions both on costs and couplings can help reach GW approximation with linear complexity in time and memory. Vayer et al. (2019) introduced sliced GW distance, reducing the complexity to $\mathcal{O}(n \log n)$. But it requires feature-based representations for projection. Vayer et al. (2020) presented the Fused GW distance that integrates both structural (GW) and feature-based (classical OT) similarity into a unified framework. Kerdoncuff et al. (2021) developed Sampled GW that reduces computational cost by sampling subsets of the cost matrices based on the current transport plan. Li et al. (2023b) proposed the Spar-GW method to approximate GW distance by leveraging a sampling strategy to construct a sparse coupling matrix. However, the accuracy of these sampling-based methods may be compromised depending on the quality of the sampling.

The aforementioned methods lack sufficient flexibility and usually suffer from a high computational burden, especially for large-scale graphs. Recent advances in graph neural networks like graph convolutional networks (GCN) (Kipf & Welling, 2016), Graph Isomorphism Networks (GIN) (Xu et al., 2018), and Graph Transformers (Shi et al., 2020; Yun et al., 2019) make it possible to enrich structural information in unsupervised graph representation learning (Sun et al., 2020; 2023; Sun & Fan, 2024). Therefore, it is natural to bridge the GW distance and GNNs (Chen et al., 2020; Vincent-Cuaz et al., 2022). We propose NeuralGW, a neural network designed to approximate the GW distance between graph pairs. The model first encodes each Graph using both GIN and a Transformer, producing two sets of node embeddings, interpreted as empirical distributions. We compute multiple Maximum Mean Discrepancy (MMD) values between these two empirical distributions to capture alignment at different kernel scales. These values are concatenated and passed through a multilayer perceptron (MLP) to approximate the ground-truth GW distance. Once trained, NeuralGW enables efficient inference without any optimization at test time. A comparison of the complexities of our NeuralGW and the baselines can be found in Section 3.4. Our contributions are as follows.

- We propose a hybrid GIN-Transformer architecture that approximates the Gromov-Wasserstein distance between graphs in a purely data-driven manner.
- We introduce an MMD layer to improve the approximation capability of NeuralGW.
- We provide an approach to predicting the transport plan of GW.
- We also theoretically analyze the generalization ability of our model.

Experiments on benchmark datasets show that our model closely approximates the true GW distance while being orders of magnitude faster. The downstream task graph clustering further verifies the effectiveness of NeuralGW. This work bridges the gap between optimal transport theory and scalable neural architectures for graph comparison. The remainder of this paper is structured as follows: Section 2 reviews background on GW distance and graph neural networks. Section 3 details our model architecture. Section 5 presents experimental results. Section 6 concludes this work.

## 2 PRELIMINARY KNOWLEDGE

### 2.1 GROMOV-WASSERSTEIN DISTANCE

The Gromov-Wasserstein (GW) distance (Mémoli, 2011; Peyré et al., 2016) provides a principled way to compare distributions supported on different metric spaces by focusing on their intrinsic relational structures. Instead of aligning points directly, GW aligns the pairwise distances within

each space, thereby comparing the underlying geometries of the domains. For two metric measure spaces $(X, d_X, \mu)$ and $(Y, d_Y, \nu)$, the squared GW distance is defined as:

$$\text{GW}^2(\mu, \nu) = \min_{\pi \in \Pi(\mu, \nu)} \sum_{i,j,k,l} |d_X(\boldsymbol{x}_i, \boldsymbol{x}_k) - d_Y(\boldsymbol{y}_j, \boldsymbol{y}_l)|^2 \pi_{ij} \pi_{kl}$$

where $\pi$ is a coupling between the two measures. This formulation naturally applies to unweighted and undirected graphs, whose adjacency matrices induce metric structures compatible with the standard GW framework. By capturing structural rather than positional similarity, GW has become a powerful tool for tasks such as graph comparison, graph matching, and more broadly, structure-aware distribution analysis.

## 2.2 Maximum Mean Discrepancy

A widely used approach for measuring the discrepancy between two probability distributions is the Maximum Mean Discrepancy (MMD) (Gretton et al., 2012). Given two distributions $P$ and $Q$, MMD measures the distance between their embeddings in a reproducing kernel Hilbert space (RKHS) $\mathcal{H}_k$ associated with a positive definite kernel k. It is defined as:

$$\text{MMD}_k^2(P, Q) = \|\mu_P - \mu_Q\|_k^2 = \mathbb{E}_{P \otimes P}[k(\boldsymbol{x}, \boldsymbol{x}')] + \mathbb{E}_{Q \otimes Q}[k(\boldsymbol{y}, \boldsymbol{y}')] - 2\mathbb{E}_{P \otimes Q}[k(\boldsymbol{x}, \boldsymbol{y})]$$

Given finite samples $\boldsymbol{X} = \{\boldsymbol{x}_i\}_{i \in [n_1]} \sim P$ and $\boldsymbol{Y} = \{\boldsymbol{y}_j\}_{j \in [n_2]} \sim Q$, an unbiased estimator of the squared is

$$\widehat{\text{MMD}}_k^2(\boldsymbol{X}, \boldsymbol{Y}) = \frac{1}{n_1(n_1-1)} \sum_{i \neq j}^{n_1} k(\boldsymbol{x}_i, \boldsymbol{x}_j) + \frac{1}{n_2(n_2-1)} \sum_{i \neq j}^{n_2} k(\boldsymbol{y}_i, \boldsymbol{y}_j) - \frac{2}{n_1 n_2} \sum_{i,j} k(\boldsymbol{x}_i, \boldsymbol{y}_j)$$

Although MMD offers significant computational advantages, it inherently assumes that the two input distributions $\boldsymbol{X}$ and $\boldsymbol{Y}$ are located in a shared ambient space. As a result, it cannot be applied directly when the domains differ in dimension or structure. This limitation motivates our proposed method, NeuralGW, which employs learned graph representations from GIN and Transformer to enable structure-aware comparison across heterogeneous domains.

## 2.3 Graph Neural Networks

Graph neural networks are designed especially for structural exploration and graph-level representation. GNNs operate by recursively aggregating local neighborhood information through message passing frameworks (Scarselli et al., 2008; Kipf & Welling, 2016). A widely studied formulation is:

$$\mathbf{h}_v^{(t)} = \text{UPDATE}^{(t)} \left( \mathbf{h}_v^{(t-1)}, \text{AGGREGATE}^{(t)} \left( \{\mathbf{h}_u^{(t-1)} : u \in \mathcal{N}(v)\} \right) \right),$$

where $\mathbf{h}_v^{(t)}$ denotes the hidden representation of node $v$ at layer $t$, and $\mathcal{N}(v)$ is its neighborhood set.

Among message-passing architectures, the Graph Isomorphism Network (GIN) (Xu et al., 2018) is theoretically notable for its expressive power. GIN replaces the aggregate function with a sum and uses an MLP for the update step:

$$\mathbf{h}_v^{(t)} = \text{MLP}^{(t)} \left( (1 + \epsilon^{(k)}) \cdot \mathbf{h}_v^{(t-1)} + \sum_{u \in \mathcal{N}(v)} \mathbf{h}_u^{(t-1)} \right),$$

where $\epsilon$ is either a learnable or fixed scalar. This design allows GIN to distinguish graph structures as powerfully as the 1-WL isomorphism test.

In contrast, Graph Transformers generalize beyond local neighborhoods using global self-attention mechanisms (Yun et al., 2019; Vaswani et al., 2017; Rampášek et al., 2022; Rong et al., 2020; Shi et al., 2020). For example, Shi et al. (2020) proposed the Graph Transformer operator:

$$\boldsymbol{h}_v^{(t)} = \boldsymbol{W}_1 \boldsymbol{h}_v^{(t-1)} + \sum_{u \in \mathcal{N}(v)} \alpha_{uv} \boldsymbol{W}_2 \boldsymbol{h}_u^{(t-1)}$$

where $\alpha_{uv} = \text{Softmax}\left( \frac{(\boldsymbol{W}_3 \boldsymbol{h}_v^{(t-1)})^T (\boldsymbol{W}_4 \boldsymbol{h}_u^{(t-1)})}{\sqrt{d}} \right)$ denotes a learned multi-head dot product attention scoring function. This attention mechanism enables modeling of long-range dependencies and global structure, which are difficult to capture with shallow message passing.

## 3 NEURALGW FRAMEWORK

### 3.1 MOTIVATION

Let $G = (V, E)$ denote a graph, where $V$ and $E$ are the vertex and edge sets, respectively. For any two graphs $G_i$ and $G_j$, we want to compute the GW distance, which is computationally expensive. To address the difficulty, we propose to learn a neural network $f$ parameterized by $\theta$, i.e.,

$$f_\theta : \mathbb{G} \times \mathbb{G} \to \mathbb{R} \tag{1}$$

on a set of graphs $\mathcal{D}_{\text{train}} = \{G_1, G_2, \ldots, G_N\}$, in which the GW distance, denoted as $\text{GWD}^*$, of every pair of graphs was computed by using a highly accurate solver of GW. We hope that $f_\theta$ can generalize well on unseen graph pairs, namely,

$$\mathbb{E}_{(G_i, G_j) \sim \mathbb{G} \times \mathbb{G}} \left[ |f_\theta(G_i, G_j) - \text{GWD}^*(G_i, G_j)| \right] \le \epsilon \tag{2}$$

where $\epsilon > 0$ is some small constant. Therefore, we propose to solve the following empirical risk minimization problem:

$$\underset{\theta}{\text{minimize}} \ \frac{2}{N(N-1)} \sum_{1 \le i < j \le N} \ell \left( f_\theta(G_i, G_j), y_{ij} \right) \tag{3}$$

where $y_{ij} = \text{GWD}^*(G_i, G_j)$ and $\ell$ denotes a loss function, e.g. $\ell_2$ loss. Note that the $N(N-1)/2$ samples are not independent due to the pair-wise formulation.

The task is non-trivial due to the following challenges:

- Graphs are non-Euclidean data, and the properties are determined by the abstract structures. It is difficult to represent each graph as a vector to facilitate the use of neural networks.

- Although graphs encode relational structures among entities, graphs originating from different domains typically vary in both semantic interpretation and scale. Such heterogeneity introduces significant challenges for cross-domain learning and generalization.

- Because of the inherent complexity, any mismatch between the model design and the data distribution can lead to substantial overfitting or underfitting.

### 3.2 MODEL ARCHITECTURE

First, we reformulate the GW distance for two graphs as

$$\text{GWD}^2(G, G') = \min_{\pi \in \Pi(\mu, \nu)} \sum_{i, j, k, l} |C_{ik}(G) - C_{jl}(G')|^2 \pi_{ij} \pi_{kl} \triangleq f(G, G') \tag{4}$$

where $C_{ik}(G)$ denotes the element $(i, k)$ of the cost matrix as a function of $G$. This means the GW distance is indeed a function, though very complex, of two graphs, which indicates the feasibility of approximating the GW distance using neural networks due to the universal approximation theorems (Hornik et al., 1989).

Here, we introduce the NeuralGW approach that is based on a GIN and a Transformer for graph representation learning. GIN is effective in learning the local structure of graphs, while GT is effective in learning the global structure of graphs. We therefore combine them to learn the node embeddings of graphs. Let $\boldsymbol{E}_i$ be the $d$-dimensional positional encoding of graph $G_i$, with $|V_i| = n_i$. We construct $\boldsymbol{E}_i$ using the singular value decomposition (SVD) on the adjacency matrix $\boldsymbol{A}_i$ of $G_i$:

$$\boldsymbol{E}_i = (\boldsymbol{u}_1, \boldsymbol{u}_2, \ldots, \boldsymbol{u}_d) \, \text{diag} \left( \sigma_1^{1/2}, \sigma_2^{1/2}, \ldots, \sigma_d^{1/2} \right) \in \mathbb{R}^{n_i \times d} \tag{5}$$

where $\boldsymbol{u}_j$ and $\sigma_j$ are the singular vector and singular values of $\boldsymbol{A}_i$, i.e., $\boldsymbol{A}_i = \boldsymbol{U} \boldsymbol{\Sigma} \boldsymbol{V}^\top$, and $\sigma_1 \ge \sigma_2 \ge \cdots \ge \sigma_{n_i}$. Note that if $n_i < d$, we get an $n_i \times n_i$ $\boldsymbol{E}_i$ first and then pad it with $d - n_i$ zero columns. Besides this SVD method, there are other positional encoding methods available. In this work, we focus on this SVD method due to its simplicity and practical effectiveness.

Let $h$ be the neural network composed of the GIN and GT, of which the parameters are denoted as $\theta$. The representations given by $h$ are denoted as $\boldsymbol{Z}_i = h_\theta(\boldsymbol{A}_i, \boldsymbol{E}_i)$. A naive method to predict GW

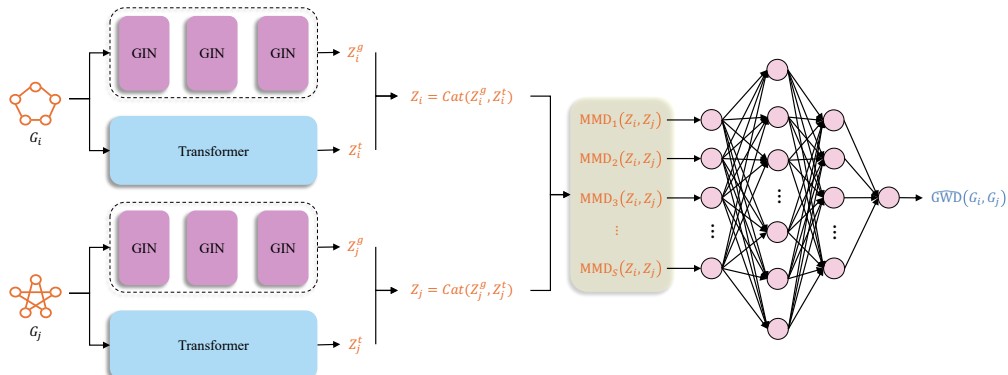

Figure 1: Architecture of NeuralGW. The GINs and Transformers are shared by the two inputs.

distance is to represent each graph as a vector by performing some pooling or readout operation (Xu et al., 2018) on $\boldsymbol{Z}_i$, e.g.,

$$\bar{\boldsymbol{z}}_i = \text{READOUT}(\boldsymbol{Z}_i) \in \mathbb{R}^d \tag{6}$$

and then use some distance metric between $\bar{\boldsymbol{z}}_i$ and $\bar{\boldsymbol{z}}_i$ to predict the GW distance, e.g.,

$$\widehat{\text{GWD}}(G_i, G_j) = \|\bar{\boldsymbol{z}}_i - \bar{\boldsymbol{z}}_j\| \tag{7}$$

We call the method NeuralGWD-Naive. Since the READOUT operation, such as sum and mean, replaces a matrix with a single vector, the information loss of the node embeddings can be significant. As a result, this naive method may not be effective enough to capture the structural information of graphs and has low expressiveness, leading to an unsatisfactory approximation for the GW distance. This will be shown in Section 5.

For each graph $G_i$, we have a set of $d$-dimensional points $\boldsymbol{Z}_i$, which can be regarded as a discrete distribution. Given that the GW distance is a distance between two sets of points, we proposed using a combination of multiple computation-efficient distances between two sets of points to predict it. In this work, we choose the MMD. Specifically, let $\alpha_{ij}^{(s)} = \text{MMD}_s(\boldsymbol{Z}_i, \boldsymbol{Z}_j)$, where $\text{MMD}_s$ denotes the MMD value based on the $s$-th kernel function $k_s(\cdot, \cdot)$, $s \in [S]$. We use the Gaussian kernel family, $k(\boldsymbol{z}, \boldsymbol{z}') = \exp(-\gamma_s \|\boldsymbol{z} - \boldsymbol{z}'\|^2)$, where $\gamma_s$ is a hyperparameter and $s \in [S]$. The prediction for GWD between $G_i$ and $G_j$ is given by

$$\widehat{\text{GWD}}_w(G_i, G_j) = g_w(\boldsymbol{\alpha}_{ij}) \tag{8}$$

where $g_w$ denotes a multilayer perceptron with parameters $w$. We let the activation function in the output layer of $g_w$ be ReLU to ensure the non-negativity of the distance measure. The architecture of our NeuralGW is in Figure 1.

Given a collection of graphs $\mathcal{D}_{\text{train}} = \{G_i\}_{i=1}^N$, we first use the standard solver from the POT package (Flamary et al., 2021)[1] to compute the ground-truth GW distances of all pairs of graphs, where the cost matrices can be the adjacency matrices or the shortest path matrices. Throughout this work, adjacency matrices are used as the cost matrices in the GW formulation. Based on the corresponding adjacency matrices $\{\boldsymbol{A}_i\}_{i=1}^N$, we obtain the positional encodings of all graphs using equation 5. Then we solve the following problem

$$\underset{\phi, w}{\text{minimize}} \ \frac{2}{N(N-1)} \sum_{1 \leq i < j \leq N} \ell\left(\widehat{\text{GWD}}_w(G_i, G_j), \text{GWD}(G_i, G_j)\right) \tag{9}$$

where $\widehat{\text{GWD}}_w(G_i, G_j) = g_w\left([\text{MMD}_1(\boldsymbol{Z}_i, \boldsymbol{Z}_j), \ldots, \text{MMD}_S(\boldsymbol{Z}_i, \boldsymbol{Z}_j)]^\top\right)$, $\boldsymbol{Z}_i = h_\phi(\boldsymbol{A}_i, \boldsymbol{E}_i)$, and $\boldsymbol{Z}_j = h_\phi(\boldsymbol{A}_j, \boldsymbol{E}_j)$. The training steps of NeuralGW are summarized in Algorithm 1, where we let $\ell$ be the quadratic loss.

Our NeuralGW indeed defines a pseudo-metric on the collection of graphs that may come from various field datasets $\{\mathcal{G}_p\}_{p=1}^P$, under some conditions.

---

[1] https://pythonot.github.io/index.html

---

**Algorithm 1** NeuralGW

---

**Require:** Collection of graphs $\mathcal{D}_{\text{train}} = \{G_1, G_2, \ldots, G_N\}$; GIN-Transformer with weight parameter $\phi$ and MLP with weight parameter set $w$

1: **for** epoch $t = 1$ to $T$ **do**
2:     Randomly select a mini-batch of graphs $\{G_i\}_{i \in \mathcal{B}}$ from $\mathcal{D}_{\text{train}}$ with replacement
3:     **for** $i = 1$ to $|\mathcal{B}|$ **do**
4:         % Extract the positional encoding of $G_i$ from its adjacency:
5:         $\boldsymbol{E}_i = \text{exact\_pe\_from\_adj}(\boldsymbol{A}_i)$ where $G_i = \mathcal{B}[i]$
6:         **for** $j = 1$ to $|\mathcal{B}|$ **do**
7:             % Extract the positional encoding of $G_j$ from its adjacency:
8:             $\boldsymbol{E}_j = \text{exact\_pe\_from\_adj}(\boldsymbol{A}_j)$ where $G_j = \mathcal{B}[j]$
9:             $\boldsymbol{Z}_i^g = \text{GIN}(\boldsymbol{E}_i, \boldsymbol{A}_i)$ and $\boldsymbol{Z}_i^t = \text{Transformer}(\boldsymbol{E}_i)$
10:           $\boldsymbol{Z}_j^g = \text{GIN}(\boldsymbol{E}_j, \boldsymbol{A}_j)$ and $\boldsymbol{Z}_j^t = \text{Transformer}(\boldsymbol{E}_j)$
11:           Concatenate $\boldsymbol{Z}_i = [\boldsymbol{Z}_i^g, \boldsymbol{Z}_i^t]$ and $\boldsymbol{Z}_j = [\boldsymbol{Z}_j^g, \boldsymbol{Z}_j^t]$
12:           Prepare for a sequence of MMDs with different kernel functions:
13:           $\{\text{MMD}_s(\boldsymbol{Z}_i, \boldsymbol{Z}_j)\}_{s=1}^S$
14:           Do feed-forward propagation: $\widehat{\text{GWD}}(G_i, G_j) = \text{MLP}(\{\text{MMD}_s(\boldsymbol{Z}_i, \boldsymbol{Z}_j)\}_{s=1}^S)$
15:           Compute the loss: $\ell(G_i, G_j) = (\widehat{\text{GWD}}(G_i, G_j) - \text{GWD}(G_i, G_j))^2$
16:         **end for**
17:     **end for**
18:     Accumulate the losses of the whole batch $\mathcal{B}$: $\ell(\mathcal{B}) = \sum_{i,j} \ell(G_i, G_j)$ and do the backward propagation untils the criterion is achieved
19: **end for**
**Ensure:** $(\phi, w)$

---

**Proposition 1.** *NeuralGW is a pseudo-metric if in the MLP the weights are nonnegative, the biases are zero, and the activation functions are ReLU.*

The proof for the proposition is deferred to Appendix A. It should be noted that the nonnegative weight assumption in the proposition is somewhat restrictive. In the experiments, we do not impose the restriction, since a practical estimator of the GW distance is not required to be a metric. On the other hand, even without the assumptions in the proposition, if both the training error and the generalization gap (see Section 3.3) are tiny, NeuralGW is almost a pseudo-metric.

**Extension: Transport Plan Prediction** We defer the details to Appendix E due to the page limit.

### 3.3 GENERALIZATION ERROR BOUND

Given $N$ i.i.d. graph samples $\mathcal{G} = \{G_i\}_{i=1}^N$ from distribution $\mathcal{D}_G$, let $\mathcal{F} = \{(G, G') \mapsto f_\theta(G, G') : \theta \in \Theta\}$ be a class of real-valued pairwise predictors. Let the target pairwise GW distance be denoted by $\text{GWD}(G, G')$ and define the squared loss class $\mathcal{L} = \{(G, G') \mapsto \ell_f(G, G') : f \in \mathcal{F}\}$ with $\ell_f(G, G') = (f(G, G') - \text{GWD}(G, G'))^2$. We assume both GWD and $f$ are bounded, *i.e.*, $\sup_{G, G' \sim \mathcal{D}_G}\{\text{GWD}(G, G'), f(G, G')\} \le M$, it holds that $0 \le \ell_f \le B = (2M)^2$ for all $\ell_f \in \mathcal{L}$. We denote the empirical risk as $U_N(\ell_f) = \frac{2}{N(N-1)} \sum_{1 \le i < j \le N} \ell_f(G_i, G_j)$ and the population risk as $R(\ell_f) = \mathbb{E}[\ell_f(G, G')]$. We then have the following generalization bound.

**Theorem 1** (Generalization bound)**.** *Let $L_{GIN}$ be the Lipschitz constant of a single GIN layer and $L_{mha}$ be the Lipschitz constant of a single Transformer layer. Assume that NeuralGW has $K_1$ GIN layers and $K_2$ Transformer layers, and $\boldsymbol{E} = [\boldsymbol{E}_1, \ldots, \boldsymbol{E}_N]$ and $\boldsymbol{A} = diag(\boldsymbol{A}_1, \ldots, \boldsymbol{A}_N)$ are the input data. For any $\delta \in (0, 1)$, with probability at least $1 - \delta$, there exists*

$$\sup_{\ell_f \in \mathcal{L}} |R(\ell_f) - U_N(\ell_f)| \le 4M\psi(1 + \log\frac{4M}{\psi}) + B\sqrt{\frac{8\log(2/\delta)}{N}} \tag{10}$$

*where $\psi = \frac{12d}{N}\sqrt{L_{MLP}C(L_{GIN}^{2K_1}\|\boldsymbol{AE}\|_F^2 + L_{mha}^{2K_2}\|\boldsymbol{E}\|_F^2)\log(2d^2)}$, and $C = 4\sqrt{\frac{2S\max_s \gamma_s}{n}}$.*

In the light of Theorem 1, let $\hat{f} = \arg\min_{f \in \mathcal{F}} U_N(\ell_f)$ be an empirical risk minimizer and $f^* = \arg\min_{f \in \mathcal{F}} R(\ell_f)$ be the global minimizer. We have $U_N(\ell_{\hat{f}}) - R(\ell_{f^*}) \le 4M\psi(1 + \log\frac{4M}{\psi}) + $

$4M^2\sqrt{8N^{-1}\log(2/\delta)}$. The theorem shows the influence of the model architecture, data property, and loss function on the gap between the training error and the expected testing error. We have the following observations: 1) The presence of $\|\boldsymbol{AE}\|_F$ indicates that when the largest spectral norm of the graphs is smaller, the bound is tighter; 2) due to the dependence of the power of $K_1$ and $K_2$, the GIN and transformer shouldn't be too deep; 3) The impact of $S$ on the bound is minor, but a large $S$ could improve the expressiveness of the model, as shown by Figure 2.

## 3.4 COMPLEXITY ANALYSIS

In NeuralGW, without loss of generality, we assume that the width and depth of GIN, Transformer, and MLP are all $\bar{d}$ and $L$ respectively, where $d < \bar{d} < n$. In the training stage, assuming that all the $N$ training graphs have $n$ nodes, the time complexity of computing the positional encodings is $\mathcal{O}(Nn^2d)$. The time complexity of computing the $S$ MMDs for any two graphs in each iteration is $\mathcal{O}(Sn^2\bar{d})$. Then the time complexity in the optimization is

Table 1: Time complexity comparison between GW, entropic GW, and NeuralGW

|  | time complexity |
| --- | --- |
| GW | $\mathcal{O}(n^3)$ |
| Entropic-GW | $\mathcal{O}(n^3)$ |
| NeuralGW | $\mathcal{O}(\bar{d}n^2)$ |

$\mathcal{O}((Bn^2\bar{d}L + B^2Sn^2\bar{d})T)$, where $T$ denotes the number of epochs of iterations and $B$ is the number of graphs sampled in each epoch. As a result, the total time complexity in the training stage is $\mathcal{O}(Nn^2d + (Nn^2\bar{d}L + N^2Sn^2\bar{d})T)$. The details of the derivation is in Appendix C.

In the testing stage, for any two graphs with nodes $n$, the time complexity is composed of the positional encoding complexity and neural network forward pass complexity, which is $\mathcal{O}(n^2d + Ln^2\bar{d} + Sn^2\bar{d})$ in total. If we treat $L$ and $S$ as constants, the time complexity of NeuralGW is $\mathcal{O}(\bar{d}n^2)$, which is lower than those of GW and Entropic-GW, as shown in Table 1. Note that the complexities of GW and Entropic-GW in the table are per-iteration complexities, where the number of required iterations could be much larger than $L$ and $S$. An empirical result can be found in Appendix D.3.

## 4 RELATED WORK

### 4.1 ACCELERATED OPTIMIZATION METHODS FOR GWD

The development of accelerated methods for computing the Gromov-Wasserstein distance is an active area of research Zhang et al. (2024). Xu et al. (2019) proposed to use proximal point algorithm (PPA) for solving the GW distance while Li et al. (2023a) introduced Bregman Alternated Projected Gradient (BAPG) method for GW computation. In a different direction, Scetbon et al. (2022) demonstrated that low-rank assumptions on costs and couplings enable a GW approximation with linear time and memory complexity. Alternatively, sampling-based strategies aim to lower computational costs: Kerdoncuff et al. (2021) introduced Sampled GW, which reduces cost by sampling subsets of the cost matrices, and Li et al. (2023b) put forward Spar-GW, which approximates the GW distance by constructing a sparse coupling matrix through sampling. The accuracy of such methods, however, is contingent upon the quality of the sampling and may consequently be compromised.

### 4.2 LEARNING-BASED APPROXIMATION METHOD FOR GWD

Many studies construct neural solvers to accelerate the approximation of discrete/continuous GW distances (Mazelet et al., 2025; Qian et al., 2024; Frogner et al., 2019). Courty et al. (2017); Klein et al. (2024) introduced a siamese architecture with a decoder to find an embedding space where the Euclidean distance mimics the GW distance. Haviv et al. (2024) presented a transformer-based autoencoder, the Wasserstein Wormhole, which embeds distributions into a latent space where Euclidean distances approximate OT distances. However, these methods still have high approximation errors. Carrasco et al. (2023) summarized five methods as representatives of solving continuous GW. Four of them, including StructuredGW (Sebbouh et al., 2024), FlowGW (Klein et al., 2023), AlignGW (Alvarez-Melis & Jaakkola, 2018), and RegGW (Uscidda et al., 2024) heavily rely on Sinkhorn acceleration in entropic GW, whereas CycleGW (Zhang et al., 2022) proposed the unbalanced bidirectional Gromov-Monge divergence. All of them are very different from our NeuralGW.

## 5 NUMERICAL RESULTS

### 5.1 IN-DOMAIN PREDICTION

We compared Sampled GW distance (Kerdoncuff et al., 2021), Spar-GW (Li et al., 2023b), and Wasserstein Wormhole (Haviv et al., 2024) with our NeuralGW and NeuralGW-Naive on four datasets in TUDataset (Morris et al., 2020) to check their capability to approximate the ground-truth GW distance [2]. Among these baselines, both Sampled GW and Spar-GW aim to accelerate the computation of GW by using a sampling technique, while Wormhole uses an attention-based encoder-decoder for the approximation of GW. We do not include DWE as a baseline since it has a 2d/3d convolution-based architecture (Courty et al., 2017). We also include $NeuralGW_{GPS}$ as an ablation study, in which the parallel GIN-Transformer architecture is replaced by the GPS module from (Rampášek et al., 2022). Likewise, for the $NeuralGW_{GPS}$-Naive variant, both the intermediate MMD layer and the final MLP module are removed. Table 2 shows the relative error of in-domain prediction of these methods on the distances of graph pairs. From Table 2, we can find that our NeuralGW is more powerful in approximating the ground-truth GW distance compared to the baselines. The results of NeuralGW-Naive demonstrate that our design of the NeuralGW architecture is necessary and rational. The results on COLLAB and REDDIT-BINARY are in Table 13 of Appendix.

Table 2: Relative error of in-domain prediction. The best result in each case is marked in bold.

|  | MUTAG | BZR | COX2 | DHFR |
|---|---|---|---|---|
| Sampled GW | 0.4527 | 0.2916 | 0.4256 | 0.3223 |
| Spar-GW (8n) | 0.7677 | 1.0872 | 1.3896 | 1.0398 |
| Spar-GW (4n) | 0.6991 | 1.1227 | 1.5138 | 1.1161 |
| Spar-GW (2n) | 0.7982 | 1.1436 | 1.4421 | 1.1104 |
| Wormhole ($\beta = 1$) | $0.5416 \pm 0.0194$ | $0.5029 \pm 0.0156$ | $0.5908 \pm 0.0304$ | $0.4909 \pm 0.0098$ |
| Wormhole ($\beta = 0$) | $0.5046 \pm 0.0171$ | $0.4142 \pm 0.0216$ | $0.3997 \pm 0.0098$ | $0.4238 \pm 0.0058$ |
| $NeuralGW_{GPS}$-Naive | $0.4943 \pm 0.0741$ | $0.4277 \pm 0.0306$ | $0.5189 \pm 0.1009$ | $0.4603 \pm 0.0674$ |
| $NeuralGW_{GPS}$ | $0.2427 \pm 0.0170$ | $0.2104 \pm 0.0024$ | $\mathbf{0.1410} \pm 0.0025$ | $0.2033 \pm 0.0182$ |
| NeuralGW-Naive | $0.3087 \pm 0.0106$ | $0.3036 \pm 0.0157$ | $0.3108 \pm 0.0179$ | $0.2575 \pm 0.0075$ |
| NeuralGW | $\mathbf{0.1414} \pm 0.0164$ | $\mathbf{0.1779} \pm 0.0445$ | $0.1466 \pm 0.0075$ | $\mathbf{0.1267} \pm 0.0077$ |

### 5.2 CROSS-DOMAIN PREDICTION

We also used multiple datasets in TUDataset (Morris et al., 2020)[3] to train our NeuralGW and NeuralGW-Naive, respectively, to check whether our model can indeed be capable of bridging the cross-domain graph worlds. The experimental details can be found in Appendix.

Table 3: Relative error of cross-domain prediction

| Trainsets
Testset | BZR + PTC_FM
MUTAG | BZR + PTC_FM
DHFR | BZR + COX2
DHFR |
|---|---|---|---|
| Sampled GW | 0.4527 | $\mathbf{0.3223}$ | 0.3223 |
| Spar-GW (8n) | 0.7677 | 1.0398 | 1.0398 |
| Spar-GW (4n) | 0.6991 | 1.1161 | 1.1161 |
| Spar-GW (2n) | 0.7982 | 1.1104 | 1.1104 |
| Wormhole ($\beta = 1$) | $0.4141 \pm 0.0063$ | $0.6737 \pm 0.0449$ | $0.7157 \pm 0.0649$ |
| Wormhole ($\beta = 0$) | $0.3978 \pm 0.0109$ | $0.5524 \pm 0.0314$ | $0.5894 \pm 0.0406$ |
| $NeuralGW_{GPS}$-Naive | $0.5112 \pm 0.1376$ | $0.5176 \pm 0.1566$ | $0.3877 \pm 0.0137$ |
| $NeuralGW_{GPS}$ | $0.3116 \pm 0.0416$ | $0.4321 \pm 0.1620$ | $0.2764 \pm 0.0186$ |
| NeuralGW-Naive | $0.3806 \pm 0.0321$ | $0.3912 \pm 0.0246$ | $0.3583 \pm 0.0319$ |
| NeuralGW | $\mathbf{0.2280} \pm 0.0158$ | $0.3439 \pm 0.0919$ | $\mathbf{0.1521} \pm 0.0479$ |

---

[2]We take the output of POT package as the ground truth, despite the fact that GW distance arises from a non-convex quadratic minimization problem.

[3]https://chrsmrrs.github.io/datasets/docs/datasets/

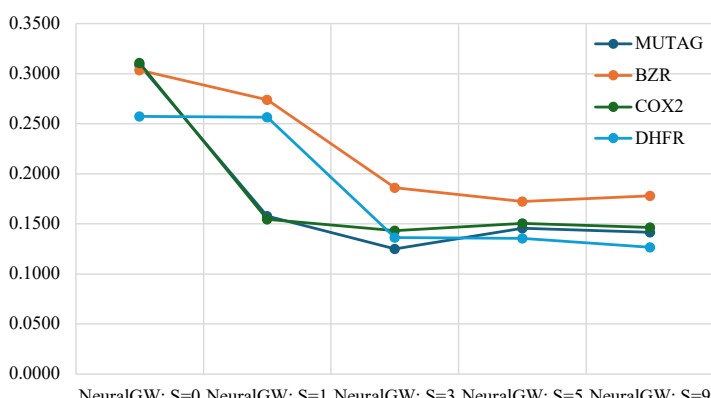

Figure 2: Sensitivity curve on the number of MMDs in the MMD layer

From the first column of Table 3, our NeuralGW can be well-trained on the combination of BZR and PTC_FM to instead approximate the GW distance matrix of cross-domain MUTAG and outperform other alternatives including NeuralGW-Naive. This is the case from two relatively heavy datasets to a simple dataset. When we choose DHFR as the target domain for prediction, our NeuralGW trained on BZR + PTC_FM cannot make a significant difference from NeuralGW-Naive, as in the second column. One reason could be that the combination of BZR and PTC_FM is as moderate as DHFR (Details for datasets are in Appendix). Luckily, as long as we substitute PTC_FM with COX2, the case immediately becomes better, i.e., our NeuralGW outperforms the other two methods again. That is, by properly choosing a combination of small datasets, our NeuralGW can be well-trained on it and help to predict the GW distance for a larger and heterogeneous dataset. Note that the results on larger datasets are in the appendix.

## 5.3 SENSITIVITY ANALYSIS

The MMD layer is a key ingredient of our NeuralGW architecture. It can potentially capture multi-scale information of difference between the latent embeddings of graph pairs. To numerically verify the necessity of this design, we compare the performance of NeuralGW with different number of MMDs on four datasets. From Figure 2, it can be observed that the MMD layer with more than a single MMD could help NeuralGW achieve better performance. The corresponding numerical results can be found in Table 9. Of course, it is also important to choose an appropriate number of MMDs for a specific dataset. Note that NeuralGW: S=0 corresponds exactly to the NeuralGW-Naive variant.

## 5.4 GRAPH CLUSTERING

Our NeuralGW has the potential for various downstream tasks where the distance matrix is required, such as graph clustering (Shi & Malik, 2000). We compared the performance of spectral cluster-ing using the GW distance and its approximations including Sampled GW, Spar-GW with different sampling rates, Wormhole GW[4] trained on PTC_FM + BZR, NeuralGW trained on PTC_FM, Neu-ralGW trained on BZR, and NeuralGW trained on PTC_FM + BZR. The results are summarized in Table 4. We see that the clustering performance of NeuralGW trained on PTC_FM or PTC_FM + BZR is close to that of GW and Spar-GW with a high sampling rate. This further demonstrated the generalization ability of NeuralGW to different domains.

---

[4]The original Wasserstein Wormhole is an attention-based encoder-decoder architecture. We use Wormhole$_{\beta=1}$ to denote the original model while using Wormhole$_{\beta=0}$ to refer to the model without an attention-based decoder.

Table 4: Graph Clustering on MUTAG. ACC, NMI, ARI, and RI denote the clustering accuracy, normalized mutual information, adjusted rand index, and rand index, which are widely used metrics for clustering. The best result of the approximation in each case is highlighted in bold.

| | ACC | NMI | ARI | RI |
|---|---|---|---|---|
| GW | 0.8457 | 0.3751 | 0.4736 | 0.7377 |
| random walk kernel | 0.7766 | **0.3082** | 0.3026 | 0.6512 |
| WL kernel | 0.7074 | 0.1360 | 0.1678 | 0.5839 |
| WL-optimal assignment kernel | 0.6809 | 0.1774 | 0.1237 | 0.5631 |
| Sampled GW | 0.7500 | 0.2393 | 0.2464 | 0.6230 |
| Spar-GW (8n) | 0.8191 | 0.2790 | 0.3977 | 0.7021 |
| Spar-GW (4n) | 0.7606 | 0.1623 | 0.2581 | 0.6339 |
| Spar-GW (2n) | 0.7872 | 0.2368 | 0.3239 | 0.6632 |
| Wormhole$_{\beta=1}$ (PTC_FM+BZR) | $0.5479 \pm 0.0274$ | $0.0038 \pm 0.0036$ | $0.0046 \pm 0.0100$ | $0.5031 \pm 0.0057$ |
| Wormhole$_{\beta=0}$ (PTC_FM+BZR) | $0.5181 \pm 0.0759$ | $0.0039 \pm 0.0058$ | $0.0082 \pm 0.0138$ | $0.5072 \pm 0.0068$ |
| NeuralGW (PTC_FM) | $\mathbf{0.8245} \pm 0.0100$ | $0.2937 \pm 0.0264$ | $\mathbf{0.4118} \pm 0.0271$ | $\mathbf{0.7092} \pm 0.0129$ |
| NeuralGW (BZR) | $0.6777 \pm 0.2017$ | $0.2030 \pm 0.1307$ | $0.2464 \pm 0.1659$ | $0.6262 \pm 0.0798$ |
| NeuralGW (PTC_FM+BZR) | $\mathbf{0.8245} \pm 0.0065$ | $0.2885 \pm 0.0163$ | $0.4101 \pm 0.0179$ | $0.7091 \pm 0.0086$ |

## 6 CONCLUSION

GW distance is powerful enough for comparing graphs, but its computation is time-consuming. We have proposed NeuralGW to approximate the GW distance to do fast structural comparisons between graphs. Note that the training of our NeuralGW only needs the adjacency matrix as required by GW distance, and thus is as applicable as GW distance. We applied the proposed NeuralGW to various experiments on some real-world datasets, which demonstrated its effectiveness and efficiency.

One limitation of this work is that we only used the SVD-based positional encodings of graphs. It is possible that including more types of encodings can improve the performance of our NeuralGW.

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

## A    PROOF FOR PROPOSITION 1 (PSEUDO METRIC)

*Proof.* To prove that our NeuralGW defines a pseudo-metric, we need to prove the following properties:

1. Positivity: $d_{\text{NeuralGW}}(G_1, G_2) \geq 0$, and $d_{\text{NeuralGW}}(G_1, G_2) = 0 \Leftarrow G_1 = G_2$;
2. Symmetry: $d_{\text{NeuralGW}}(G_1, G_2) = d_{\text{NeuralGW}}(G_2, G_1)$;
3. Triangle inequality: $d_{\text{NeuralGW}}(G_1, G_2) \leq d_{\text{NeuralGW}}(G_1, G_3) + d_{\text{NeuralGW}}(G_2, G_3)$, under some assumptions;

We first claim that NeuralGW is permutation invariant by design. This follows from two key observations:

1. Under the graph learning setting, GIN and Transformer layers are permutation equivariant, meaning that permuting the input node order results in an identical permutation of the output node representations;

2. MMD is permutation invariant, as it depends only on pairwise similarities between sets and is therefore unaffected by the ordering of elements

Since the MMD module serves as a set-level readout layer, it produces graph-level representations that are themselves permutation invariant. The subsequent MLP operates on these graph-level embeddings and, being applied after an invariant readout, cannot break permutation invariance. Therefore, the overall NeuralGW architecture is permutation invariant.

Now, we show that the properties hold under the assumptions made in the proposition.

(1) Since the activation function in the output layer is ReLU, we have $d_{\text{NeuralGW}}(G_1, G_2) \geq 0$. Moreover, for two identical graphs $G1 = G_2$, their embeddings produced by the GIN and Transformer layers will coincide. Consequently, the

$$\alpha^{(s)} = \text{MMD}_s(\mathbf{Z}_1, \mathbf{Z}_2) = 0, \quad \forall s = 1, 2, \ldots, S$$

Since the activation functions in the MLP are ReLu and the biases are all zero, we have $d_{\text{NeuralGW}}(G_1, G_2) = g_w(\boldsymbol{\alpha}) = 0$. Therefore, Property 1 holds.

(2) Property 2 (Symmetry) immediately follows from that fact that MMDs in the MMD layer are symmetric.

(3) For convenience, let $\alpha_{i,j}^{(s)} = \text{MMD}_s(\mathbf{Z}_i, \mathbf{Z}_j)$. Since MMD satisfies the triangle inequality, we have

$$\alpha_{1,2}^{(s)} \leq \alpha_{1,3}^{(s)} + \alpha_{2,3}^{(s)} \tag{11}$$

Multiplying both sides by $w_s \geq 0$ and summarizing over $s = 1, 2, \ldots, S$ yields

$$\sum_{s=1}^{S} w_s \alpha_{1,2}^{(s)} \leq \sum_{s=1}^{S} w_s \alpha_{1,3}^{(s)} + \sum_{s=1}^{S} w_s \alpha_{2,3}^{(s)} \tag{12}$$

Performing the ReLU activation function on the left-hand side of the above inequality, we have

$$
\begin{aligned}
& \text{ReLU}\left(\sum_{s=1}^{S} w_s \alpha_{1,2}^{(s)}\right) \\
&= \sum_{s=1}^{S} w_s \alpha_{1,2}^{(s)} \\
&\leq \sum_{s=1}^{S} w_s \alpha_{1,3}^{(s)} + \sum_{s=1}^{S} w_s \alpha_{2,3}^{(s)} \\
&= \text{ReLU}\left(\sum_{s=1}^{S} w_s \alpha_{1,3}^{(s)}\right) + \text{ReLU}\left(\sum_{s=1}^{S} w_s \alpha_{2,3}^{(s)}\right)
\end{aligned}
\tag{13}
$$

This means the output distance of each hidden node in each hidden layer of the MLP satisfies the triangle inequality. By composition and recursion, we conclude that

$$d_{\text{NeuralGW}}(G_1, G_2) \leq d_{\text{NeuralGW}}(G_1, G_3) + d_{\text{NeuralGW}}(G_2, G_3) \tag{14}$$

$\square$

## B  PROOF OF GENERALIZATION BOUND IN THEOREM 1

Before proving the theorem, we show some technical lemmas.

**Lemma 1.** *(Symmetrization/decoupling for U-statistics) For U-statistic of order 2 and class $\mathcal{L}$,*

$$\mathbb{E}[\sup_{\ell_f \in \mathcal{L}} (U_N(\ell_f) - R(\ell_f))] \leq 2\mathbb{E}[\sup_{\ell_f \in \mathcal{L}} \frac{1}{\lfloor N/2 \rfloor} \sum_{i=1}^{\lfloor N/2 \rfloor} \sigma_i \ell_f(G_i, G_{i+\lfloor N/2 \rfloor})]$$

*where $\{\sigma_i\}_{i=1}^{\lfloor N/2 \rfloor}$ are i.i.d. Rademacher signs independent of the sample.*

*Proof of Lemma 1.* **Setup and notation** Let entries of $X = (X_1, X_2, \ldots, X_N)$ be *i.i.d.* random variables taking values in some space $\mathcal{X}$. Let $\ell_f : \mathcal{X} \times \mathcal{X} \to \mathbb{R}$ be a symmetric kernel (*i.e.*, $\ell_f(x, y) = \ell_f(y, x)$). The U-statistic of order 2 is

$$U_N(\ell_f) = \frac{2}{N(N-1)} \sum_{1 \leq i < j \leq N} \ell_f(X_i, X_j).$$

Denote $m = \lfloor N/2 \rfloor$ (For simplicity, assume $N$ is even so $m = n/2$; nothing essential will change otherwise.)

We want to relate $U_N(\ell_f)$ to averages of independent pair sums (or to Rademacher symmetrized sums) via a combinatorial permutation argument.

### 1) **The permutation/pairing representation**

Consider the set $\mathfrak{S}_n$ of all permutations $\pi$ of $[n]$. For each permutation $\pi$, define the paired-sum statistic

$$S_\pi(\ell_f) = \frac{1}{m} \sum_{k=1}^m \ell_f(X_{\pi(2k-1)}, X_{\pi(2k)}).$$

Then, Hoeffding, 1963 claims that

$$U_N(\ell_f) = \frac{1}{|\mathfrak{S}_N|} \sum_{\pi \in \mathfrak{S}_N} S_\pi(\ell_f)$$

*i.e.*, the U-statistic is the average, over all permutations, of the paired-sum statistic.

### 2) **Symmetrization for a fixed permutation**

Fix a permutation $\pi$. Consider the centered quantity $S_\pi(h) - \mathbb{E}[S_\pi(\ell_f)]$. We can symmetrize this difference using an independent sample $X' = (X'_1, X'_2, \ldots, X'_N)$ (*i.i.d.* copies of $X_i$) and Rademacher signs. A standard inequality is

$$\mathbb{E}[\sup_{\ell_f \in \mathcal{L}} |S_\pi(\ell_f) - \mathbb{E}S_\pi(\ell_f)|] \leq \mathbb{E}_{X,X'}\mathbb{E}_\sigma[\sup_{\ell_f \in \mathcal{L}} |\frac{1}{m} \sum_{i=1}^m \sigma_k(\ell_f(X_{\pi(2k-1)}, X_{\pi(2k)}) - \ell_f(X'_{\pi(2k-1)}, X'_{\pi(2k)}))|]$$

where $\sigma = (\sigma_1, \ldots, \sigma_m)$ are *i.i.d.* Rademacher signs, independent of $X, X'$.

Since $|a - b| \leq |a| + |b|$ and the two halves are identically distributed, one has

$$\mathbb{E}[\sup_{\ell_f \in \mathcal{L}} |S_\pi(\ell_f) - \mathbb{E}S_\pi(\ell_f)|] \leq 2\mathbb{E}_X\mathbb{E}_\sigma[\sup_{\ell_f \in \mathcal{L}} |\frac{1}{m} \sum_{i=1}^m \sigma_k\ell_f(X_{\pi(2k-1)}, X_{\pi(2k)})|]$$

So for each fixed $\pi$, the deviation of $S_\pi$ is controlled by a Rademacher average over the $m$ paired terms.

### 3) **Average over permutations to get the bound for U-statistic**

Since $U_N(\ell_f)$ is the average of $S_\pi(\ell_f)$ over permutations, one has

$$U_N(\ell_f) - \mathbb{E}[U_N(\ell_f)] = \frac{1}{|\mathfrak{S}_N|} \sum_{\pi \in \mathfrak{S}_N} (S_\pi(\ell_f) - \mathbb{E}[S_\pi(\ell_f)])$$

By convexity of the sup and Jensen (or simply triangular inequality for expectation of sup and averages), we get

$$\mathbb{E}[\sup_{\ell_f \in \mathcal{L}} |U_N(\ell_f) - \mathbb{E}[U_N(\ell_f)]|] \leq \frac{1}{|\mathfrak{S}_N|} \sum_{\pi \in \mathfrak{S}_N} \mathbb{E}[\sup_{\ell_f \in \mathcal{L}} |S_\pi(\ell_f) - \mathbb{E}[S_\pi(\ell_f)]|].$$

(**move sup inside the permutation average**)

Combining with the bound for each $\pi$ from Part 2) yields

$$\mathbb{E}[\sup_{\ell_f \in \mathcal{L}} |U_N(\ell_f) - \mathbb{E}[U_N(\ell_f)]|] \leq \frac{2}{|\mathfrak{S}_N|} \sum_{\pi \in \mathfrak{S}_N} \mathbb{E}_X\mathbb{E}_\sigma[\sup_{\ell_f \in \mathcal{L}} |\frac{1}{m} \sum_{i=1}^m \sigma_k\ell_f(X_{\pi(2k-1)}, X_{\pi(2k)})|]$$

$$= 2\mathbb{E}_X\mathbb{E}_\sigma[\sup_{\ell_f \in \mathcal{L}} |\frac{1}{m} \sum_{i=1}^m \sigma_k\ell_f(X_{2k-1}, X_{2k})|]$$

where the last identity holds because the distribution of the paired terms $(X_{\pi(2k-1)}, X_{\pi(2k)})$ is the same for every $\pi$ (*i.e.*, permuting indices does not change joint distribution). In other words, the

permutation average collapses to the expected Rademacher average over one arbitrary fixed pairing of the sample into $m$ disjoint pairs (because all pairings are exchangeable).

Therefore, the final symmetrization/decoupling inequality for U-statistic of order 2 is typically stated as

$$\mathbb{E}[\sup_{\ell_f \in \mathcal{L}} |U_N(\ell_f) - \mathbb{E}[U_N(\ell_f)]|] \leq 2\mathbb{E}_X \mathbb{E}_\sigma [\sup_{\ell_f \in \mathcal{L}} |\frac{1}{m} \sum_{i=1}^{m} \sigma_k \ell_f(X_{2k-1}, X_{2k})|]$$

$\square$

**Lemma 2** (Rademacher contraction for squared loss)**.** *Asumme that $|f| \leq M$ and $|t| \leq M$, the squared loss $\ell_f = (f - t)^2$ is L-Lipschitz in $f$ with the constant $L = 2M$, which gives*

$$\mathfrak{R}_N(\mathcal{L}) \leq 2M\mathfrak{R}_N(\mathcal{F})$$

*where $\mathfrak{R}_N(\mathcal{L}) = \mathbb{E}[\sup_{\ell_f \in \mathcal{L}} \frac{1}{\lfloor N/2 \rfloor} \sum_{i=1}^{\lfloor N/2 \rfloor} \sigma_i \ell_f(G_i, G_{i+\lfloor N/2 \rfloor})]$ is the empirical Rademcher complexity of a loss class, and $\mathfrak{R}_N(\mathcal{F}) = \mathbb{E}_\sigma[\sup_{f \in \mathcal{F}} \frac{1}{N} \sum_{i=1}^{N} \sigma_i f(X_i)]$ is the empirical Rademacher complexity of the function class $\mathcal{F}$ for $f \in \mathcal{F}$.*

*Proof.* This is the standard contraction lemma (Ledoux, 1997) applied to the pairing: for each paired index $i$, the mapping $f \mapsto (f - t(G_i, G_{i+\lfloor N/2 \rfloor}))^2$ is $2M$-Lipschitz on $[-M, M]$. The contraction lemma states that Rademacher averages of Lipschitz compositions are bounded by the Lipschitz constant times the Rademacher averages of the inner class. Thus, the inequality follows immediately.
$\square$

**Lemma 3** (Bounded-differences for supremum of U-statistic deviation)**.** *Define $F(G_1, \ldots, G_N) = \sum_{\ell_f \in \mathcal{L}} (U_N(\ell_f) - R(\ell_f))$ with the property that changing a single sample $G_i$ to an arbitrary $G_i'$ changes F by at most $\frac{4B}{N}$. Then, by McDiarmid et al. (1989)'s inquality, for any $\varepsilon > 0$,*

$$Pr(F - \mathbb{E}[F] \geq \varepsilon) \leq \exp(\frac{N\varepsilon^2}{8B^2}).$$

*Proof.* A single sample $G_i$ appears in exactly $N - 1$ unordered pairs $(G_i, G_j)$ or $(G_j, G_i)$. The U-statistic normalization gives each unordered pair the weight $\frac{1}{\binom{N}{2}} = \frac{2}{N(N-1)}$. Replacing $G_i$ can change each of those $N - 1$ terms by at most $B$. So the total change in $U_N(\ell_f)$ for any fixed $\ell_f$ is at most $(N - 1) \cdot B \cdot \frac{2}{N(N-1)} = \frac{2B}{N}$. Thus, $|U_N(\ell_f; G_{1:N}) - U_N(\ell_f; (G_{1:N} \backslash \{G_i\}) \cup \{G_i'\})| \leq \frac{2B}{N}$.

Now, $F$ is a supremum difference $U_N(\ell_f) - R(\ell_f)$. Replacing $G_i$ affects the first term by at most $\frac{2B}{N}$ and the second term $R(\ell_f)$ is unchanged (population expectation is independent of the sample). When we compare two suprema (before and after change), the triangle inequality gives that $F$ can change by at most twice that, *i.e.*, at most $\frac{4B}{N}$. Thus, bounded-difference constant $c_i \leq \frac{4B}{N}$.

Applying McDiarmid et al. (1989)'s inequality with $\sum_i c_i^2 = N \cdot (\frac{4B}{N})^2 = \frac{16B^2}{N}$ yields

$$\Pr(F - \mathbb{E}[F] \geq \varepsilon) \leq \exp(-\frac{2\varepsilon^2}{\sum_i c_i^2}) = \exp(-\frac{2\varepsilon^2}{16B^2/N}) = \exp(-\frac{N\varepsilon^2}{8B^2}).$$

$\square$

We want to derive a high-probability upper bound on

$$\sup_{\ell_f \in \mathcal{L}} |R(\ell_f) - U_N(\ell_f)| = \max\{\sup_{\ell_f}(R(\ell_f) - U_N(\ell_f)), \sup_{\ell_f}(U_N(\ell_f) - R(\ell_f))\}$$

It suffices to bound $F^+ = \sup_{\ell_f}(U_N(\ell_f) - R(\ell_f))$ since the other side is symmetric. We will bound $\mathbb{E}[F^+]$ and then upgrade the expectation bound to a high-probability statement via McDiarmid et al. (1989). The same bound applies to the absolute sup via union (factor 2) or by applying it twice.

By Lemma 1 (Symmetrization for U-statistics),

$$\mathbb{E}[F^+] = \mathbb{E}[\sup_{\ell_f}(U_N(\ell_f) - R(\ell_f))] \leq 2\mathbb{E}[\sup_{\ell_f \in \mathcal{L}} \frac{1}{\lfloor N/2 \rfloor} \sum_{i=1}^{\lfloor N/2 \rfloor} \sigma_i \ell_f(G_i, G_{i+\lfloor N/2 \rfloor})]$$

Define the right-hand side quantity (with explicit normalization) as the pairwise empirical Rademacher complexity

$$\mathfrak{R}_N(\mathcal{L}) = \mathbb{E}_{G,\sigma}[\sup_{\ell_f \in \mathcal{L}} \frac{1}{\lfloor N/2 \rfloor} \sum_{i=1}^{\lfloor N/2 \rfloor} \sigma_i \ell_f(G_i, G_{i+\lfloor N/2 \rfloor})],$$

where $\{\sigma_i\}_{i=1}^{\lfloor N/2 \rfloor}$ are i.i.d. Rademacher signs independent of the sample. One immediately has $\mathbb{E}[F^+] \leq 2\mathfrak{R}_N(\mathcal{L})$. Since $\mathcal{L}$ is the squared-loss class induced by $\mathcal{F}$, Lemma 2 gives $\mathfrak{R}_N(\mathcal{L}) \leq 2M\mathfrak{R}_N(\mathcal{F})$.

We are now able to convert the expectation bound to a high-probability bound. By Lemma 3 (McDiarmid et al. (1989)'s inequality with the bounded-difference constant $\frac{4B}{N}$), for any $\varepsilon > 0$,

$$\Pr(F^+ - \mathbb{E}[F^+] \geq \varepsilon) \leq \exp(-\frac{N\varepsilon^2}{8B^2}).$$

Setting the right-hand side to $\frac{\delta}{2}$, one has $\varepsilon = B\sqrt{\frac{8\log(2/\delta)}{N}}$. Thus, with probability at least $1 - \delta/2$,

$$F^+ \leq \mathbb{E}[F^+] + B\sqrt{\frac{8\log(2/\delta)}{N}}.$$

The symmetric bound can be derived for $F^- = \sup_{\ell_f}(R(\ell_f) - U_N(\ell_f))$ independently. Then, by union bound with probability at least $1 - \delta$,

$$\sup_{\ell_f \in \mathcal{L}} |R(\ell_f) - U_N(\ell_f)| \leq \mathbb{E}[F^+] + B\sqrt{\frac{8\log(2/\delta)}{N}}.$$

Here, we absorbed the factor 2 into the $\log(2/\delta)$ term.

Consequently, we have

$$\sup_{\ell_f \in \mathcal{L}} |R(\ell_f) - U_N(\ell_f)| \leq 4M\mathfrak{R}_N(\mathcal{F}) + B\sqrt{\frac{8\log(2/\delta)}{N}}$$

with probability at least $1 - \delta$.

Let $\hat{f} = \arg\min_f U_N(\ell_f)$ and $f^* = \arg\min_f R(\ell_f)$. Then, by standard argument in empirical risk minimization, $U_N(\ell_{\hat{f}}) - R(\ell_{f^*}) \leq \sup_f |R(\ell_f) - U_N(\ell_f)|$. Combing with the bound above, with the probability at least $1 - \delta$,

$$U_N(\ell_{\hat{f}}) - R(\ell_{f^*}) \leq 4M\mathfrak{R}_N(\mathcal{F}) + B\sqrt{\frac{8\log(2/\delta)}{N}}$$

Now, it's time to discuss how to estimate an upper bound on the Rademacher complexity $\mathfrak{R}_N(\mathcal{F})$. Let $\boldsymbol{E} = [\boldsymbol{E}_1; \cdots; \boldsymbol{E}_N]$ be the concatenation of the positional embeddings and $\boldsymbol{A} = \text{diag}(\boldsymbol{A}_1, \cdots, \boldsymbol{A}_N)$ be the block-diagonal matrix. Consider a big graph $G_{\text{cat}} = (\boldsymbol{E}, \boldsymbol{A})$ as the input of GIN submodule and $\boldsymbol{E}$ as the input of Transformer submodule, the covering number of $(\boldsymbol{E}, \boldsymbol{A})$ would be $(\|\boldsymbol{A}\boldsymbol{E}\|_F^2 \frac{d^2}{\varepsilon^2})\log(2d^2)$ while the covering number of $\boldsymbol{E}$ is $(\|\boldsymbol{E}\|_F^2 \frac{d^2}{\varepsilon^2})\log(2d^2)$, according to the Lemma 3.2 in Bartlett et al. (2017).

Assume the proposed NeuralGW have $K_1$ layers of GIN, each with Lipschitz constant $L_{\text{GIN}}$ and $K_2$ Transformer encoder layers, each with Lipschitz constant $L_{\text{mha}}$. Thus, the covering number of the GIN submodule will be $(L_{\text{GIN}}^{2K_1}\|\boldsymbol{A}\boldsymbol{E}\|_F^2 \frac{d^2}{\varepsilon^2})\log(2d^2)$ while the covering number of Transformer

submodule is $(L_{\mathrm{mha}}^{2K_2}\|\boldsymbol{E}\|_F^2 \frac{d^2}{\varepsilon^2})\log(2d^2)$. Taking into account the paralle architecture of NeuralGW, the total covering number would be $C(L_{\mathrm{GIN}}^{2K_1}\|\boldsymbol{AE}\|_F^2 + L_{\mathrm{mha}}^{2K_2}\|\boldsymbol{E}\|_F^2)\frac{d^2}{\varepsilon^2}\log(2d^2)$, where $C$ is the Lipschitz constant of the MMDs and the final MLP. Specifically, suppose the MLP has $V$ layers, then

$$C = 4\sqrt{\frac{2S\max_s \gamma_s}{n}}\rho^V \prod_{v=1}^{V}\|\mathbf{W}_v\|_2 \triangleq 4\sqrt{\frac{2S\max_s \gamma_s}{n}}L_{MLP} \tag{15}$$

where $\mathbf{W}_v$ is the weight matrix of layer $v$ of the MLP and $\rho$ is the Lipschitz constant of the activation functions. Then, By Lemma A.5 of Bartlett et al. (2017), Rademacher complexity of $\mathcal{F}$ will be upper-bounded as

$$\mathfrak{R}_N(\mathcal{F}) \leq \inf_{\alpha>0}\left(\frac{4\alpha\gamma}{\sqrt{N}} + \frac{12}{N}\int_{\gamma\alpha}^{\gamma\sqrt{N}}\sqrt{\log\mathcal{N}(\varepsilon,\mathcal{F},\rho)}d\varepsilon\right)$$

where $\log\mathcal{N}(\varepsilon,\mathcal{F},\rho) = C(L_{\mathrm{GIN}}^{2K_1}\|\boldsymbol{AE}\|_F^2 + L_{\mathrm{mha}}^{2K_2}\|\boldsymbol{E}\|_F^2)\frac{d^2}{\varepsilon^2}\log(2d^2)$.

Since $\log\mathcal{N}(\varepsilon,\mathcal{F},\rho) = C(L_{\mathrm{GIN}}^{2K_1}\|\boldsymbol{AE}\|_F^2 + L_{\mathrm{mha}}^{2K_2}\|\boldsymbol{E}\|_F^2)\frac{d^2}{\varepsilon^2}\log(2d^2)$, we have by Lemma A.5 of Bartlett et al. (2017)

$$\mathfrak{R}_N(\mathcal{F})$$

$$\leq \inf_{\alpha>0}\left(\frac{4\alpha M}{\sqrt{N}} + \frac{12}{N}\int_{M\alpha}^{M\sqrt{N}}\sqrt{\log\mathcal{N}(\varepsilon,\mathcal{F},\rho)}d\varepsilon\right)$$

$$= \inf_{\alpha>0}\left(\frac{4\alpha M}{\sqrt{N}} + \frac{12}{N}\int_{M\alpha}^{M\sqrt{N}}\sqrt{C(L_{\mathrm{GIN}}^{2K_1}\|\boldsymbol{AE}\|_F^2 + L_{\mathrm{mha}}^{2K_2}\|\boldsymbol{E}\|_F^2)\frac{d^2}{\varepsilon^2}\log(2d^2)}d\varepsilon\right)$$

$$= \inf_{\alpha>0}\left(\frac{4\alpha M}{\sqrt{N}} + \frac{12d}{N}\sqrt{C(L_{\mathrm{GIN}}^{2K_1}\|\boldsymbol{AE}\|_F^2 + L_{\mathrm{mha}}^{2K_2}\|\boldsymbol{E}\|_F^2)\log(2d^2)}\int_{M\alpha}^{M\sqrt{N}}\frac{1}{\varepsilon}d\varepsilon\right)$$

$$= \inf_{\alpha>0}\left(\frac{4\alpha M}{\sqrt{N}} + \frac{12d}{N}\sqrt{C(L_{\mathrm{GIN}}^{2K_1}\|\boldsymbol{AE}\|_F^2 + L_{\mathrm{mha}}^{2K_2}\|\boldsymbol{E}\|_F^2)\log(2d^2)}\log\frac{\sqrt{N}}{\alpha}\right)$$

Let $\psi = \frac{12d}{N}\sqrt{C(L_{\mathrm{GIN}}^{2K_1}\|\boldsymbol{AE}\|_F^2 + L_{\mathrm{mha}}^{2K_2}\|\boldsymbol{E}\|_F^2)\log(2d^2)}$ and define $h(\alpha) = \frac{4M}{\sqrt{N}}\alpha + \psi\log\frac{\sqrt{N}}{\alpha}$. Setting $\frac{dh}{d\alpha} = 0$, one has the stationary point $\alpha = \frac{\psi\sqrt{N}}{4M}$. Substituting it back to the upper bound, we get

$$\mathfrak{R}_N(\mathcal{F}) \leq \psi(1 + \log\frac{4M}{\psi})$$

Finally, with probability at least $1 - \delta$, we have

$$\sup_{\ell_f \in \mathcal{L}}|R(\ell_f) - U_N(\ell_f)| \leq 4M\psi(1 + \log\frac{4M}{\psi}) + B\sqrt{\frac{8\log(2/\delta)}{N}}$$

where $\psi = \frac{12d}{N}\sqrt{C(L_{\mathrm{GIN}}^{2K_1}\|\boldsymbol{AE}\|_F^2 + L_{\mathrm{mha}}^{2K_2}\|\boldsymbol{E}\|_F^2)\log(2d^2)}$.

## C  DETAILS OF COMPLEXITY ANALYSIS

We assumed that the width and depth of GIN, Transformer, and MLP are all $\bar{d}$ and $L$, respectively.

- For MLP, to process one graph, the per-layer complexity is $\mathcal{O}(n\bar{d}^2)$.
- For GIN, to process one graph, the per-layer complexity is $\mathcal{O}(n^2\bar{d} + n\bar{d}^2)$, due to the neighbor aggregation operation and feedforward operation.
- For Transformer, to process one graph, the per-layer complexity is $\mathcal{O}(n\bar{d}^2 + n^2\bar{d} + n^2\bar{d})$, where the three terms are due to weight matrix multiplications, the construction of the attention map, and the multiplication with the attention map, respectively.

- The complexity of gradient calculations is about two times that of the forward pass.

Summing up these terms and multiplying with $L$, we have $\mathcal{O}((n\bar{d}^2 + n^2\bar{d})L)$ for processing one graph in one iteration. By assuming $\bar{d} \leq n$, the complexity is simplified to $\mathcal{O}(n^2\bar{d}L)$. For $N$ graphs and $T$ iterations, as well as the complexity for all position encoders and the MMD layer, we have a complexity of $\mathcal{O}\left(Nn^2\bar{d} + \left(Nn^2\bar{d}L + N^2Sn^2\bar{d}\right)T\right)$. We added these details to the appendix.

# D    EXPERIMENTAL DETAILS

## D.1    DATASETS, SETTINGS, AND METRICS

We evaluate the generalization ability of NeuralGW through two standard settings.

- **In-domain prediction** involves splitting a single dataset into training and test subsets, allowing the model to learn and be evaluated with the same domain.
- **Cross-domain prediction**, in contrast, train the model on one or multiple source datasets and tests it on a previously unseen target dataset, making it a more challenging scenario. Additionally, under the cross-domain setting, we consider a downstream graph clustering task, which graphs are grouped based on the predicted distance matrix. The quality of the clustering results provides further insight into the cross-domain generalization capability of NeuralGW.

**Summary of datasets**    We test our method on the datasets from the TUDatasets (Morris et al., 2020) as described in Table 5. These datasets are from chemistry, bioinformatics, and social networks. For both in-domain and cross-domain sections, we evenly choose them for detailed discussion.

Table 5: Statistics of datasets

| Dataset | Type | Class | Avg. nodes | Avg. edges | Num of graphs |
|---|---|---|---|---|---|
| MUTAG | molecules | 2 | 17.93 | 19.79 | 188 |
| BZR | molecules | 2 | 35.75 | 38.36 | 405 |
| COX2 | molecules | 2 | 41.22 | 43.45 | 467 |
| DHFR | molecules | 2 | 42.43 | 44.54 | 756 |
| PTC_FM | molecules | 2 | 14.11 | 14.48 | 349 |
| DD | bioinformatics | 2 | 284.32 | 715.66 | 1178 |
| ENZYMES | bioinformatics | 6 | 32.63 | 62.14 | 600 |
| COLLAB | social networks | 3 | 74.49 | 2457.78 | 5000 |
| REDDIT-BINARY | social networks | 2 | 429.63 | 497.75 | 2000 |

**Baselines**    We include classical optimization-based methods and neural-based methods as our baselines. They are summarized as follows.

**Settings**    We implemented all of the numerical experiments with a fixed architecture of NeuralGW that has a single GIN layer with 1024 hidden units, a single Transformer with 4 heads, and 3-layer MLP. We used AdamW optimizer with the learning rate 0.0001. In the MMD layer, $S = 9$ is used across all of our experiments. In this work, we report the average and standard deviation of 5 random trials for all NN-based methods. Regarding the selection of kernel parameters $\gamma_1, \gamma_2, \ldots, \gamma_S$, we let $\gamma_s = \gamma s^2$, $s = 1, 2, \ldots, S$, where $\gamma$ is a learnable scale parameter initialized at 0.001. That means, for the 10 MMDs, we let the kernel parameters be $[\gamma \times 1^2, \gamma \times 2^2, \ldots, \gamma \times 10^2]$, respectively. This setting has been used in all experiments.

**Relative error of approximation on distance matrix**    Given the ground-truth Gromov-Wasserstein distance matrix $\boldsymbol{D}_{\mathrm{GW}}$, the relative residual error of an approximation distance matrix $\boldsymbol{D}_{\mathrm{pred}}$ is defined as

$$\mathcal{E}_{\boldsymbol{D}_{\mathrm{GW}}}(\boldsymbol{D}_{\mathrm{Pred}}) = \|\boldsymbol{D}_{\mathrm{Pred}} - \boldsymbol{D}_{\mathrm{GW}}\|_F / \|\boldsymbol{D}_{\mathrm{GW}}\|_F$$

Table 6: Baselines

| Method | Literature | Code source | Setting |
|---|---|---|---|
| Random walk kernel | Vishwanathan et al. (2010) | GraKel | Default |
| Weisfeiler-Lehman (WL) kernel | Shervashidze et al. (2011) | GraKel | n_iter=5 |
| WL optimal assignment kernel | Kriege et al. (2016) | GraKel | n_iter=5 |
| Gromov-Wasserstein (GW) | Mémoli (2011) | POT | Default as Groundtruth |
| Sampled GW | Kerdoncuff et al. (2021) | POT | nb_samples_grad=100 |
| Spar-GW (8n) | Li et al. (2023b) | Official code | n_subsample = 8n |
| Spar-GW (4n) | Li et al. (2023b) | Official code | n_subsample = 4n |
| Spar-GW (2n) | Li et al. (2023b) | Official code | n_subsample = 2n |
| Wormhole$_{\beta=1}$ | Haviv et al. (2024) | Official code | Original model |
| Wormhole$_{\beta=0}$ | Haviv et al. (2024) | Official code | Original model without the decoder |

for which $D_{\text{Pred}}$ in one of $\{D_{\text{Sampled GW}}, D_{\text{Spar-GW}}, D_{\text{NeuralGW-Naive}}, D_{\text{NeuralGW}}\}$. A lower $\mathcal{E}_{D_{\text{GW}}}$ indicates a better performance in predicting the GW distance.

For in-domain prediction, each dataset is split into training, validation, and test subsets with proportions 64:16:20. For cross-domain prediction, the training set consists of all graphs from the source datasets while validation and test sets are obtained by splitting the target set in a 20:80 ratio. All reported errors are computed on unseen test pairs.

Neural networks can easily memorize the training data, especially when they have high capacity. In our cross-domain prediction setting, NeuralGW tends to fit the source datasets perfectly, which may harm its ability to generalize to the target dataset. A validation set is therefore necessary to monitor and prevent this degradation in cross-domain generalization.

If we used a validation set taken from a small subset of the source dataset, early stopping would only protect generalization within the source domain. But we have no guarantee that this reflects generalization to the target domain, whose distribution is different and whose deviation from the source is unknown.

To properly track cross-domain generalization, we select a validation subset from the target dataset itself and evaluate NeuralGW's performance on this subset during training. When this validation performance stops improving, we stop training. Since both the validation set and the test set come from the same target distribution, improving performance on the validation set is a reliable indicator of improving performance on the test set.

For this reason, the validation set were drawn from the target dataset, not the source dataset.

### D.2 COMPARISON OF NEURALGW WITH ENTROPIC GW

Zhang et al. (2024) focus on the acceleration of Entropic GW by proposing a fast gradient computation method, which is incompatible with our topic to some extent. In this sense, we did not include it in the benchmark. The distance given by entropic GW is much larger than GWD if the entropic regularizer is not small enough. Here, we show some results in the following table, where the Entropic regularizer is set to 0.3, 0.1, or 0.01. We see that the error is much higher than that of our NeuralGW. Using a smaller regularization hyperparameter may reduce the error but often leads to unstable or very slow optimization. By the way, it is quite difficult to determine the best regularization hyperparameter for the trade-off between accuracy and efficiency.

Table 7: Average relative error of in-domain prediction

| | MUTAG | BZR | COX2 | DHFR |
|---|---|---|---|---|
| Entropic GW$_{\epsilon=0.3}$ | 1.7234 | 1.7383 | 2.2125 | 1.7303 |
| Entropic GW$_{\epsilon=0.1}$ | 1.1008 | 1.5939 | 1.9806 | 1.5739 |
| Entropic GW$_{\epsilon=0.01}$ | 0.2124 | 1.0613 | 1.9609 | 1.4273 |
| NeuralGW | $0.1414 \pm 0.0164$ | $0.1779 \pm 0.0445$ | $0.1466 \pm 0.0075$ | $0.1267 \pm 0.0077$ |

### D.3 EMPIRICAL RESULT ON TIME COMPLEXITY ANALYSIS

We use the subset of the COLLAB dataset including 10 graphs with $300 \sim 306$ nodes to compare the cost time that different methods have to spend for computing the pairwise distance matrix of 10 graphs. From Table 8, it can be observed that our proposed NeuralGW has a lower time complexity.

Table 8: Cost time for constructing distance matrix of graphs in COLLAB

| Method | GW | Entropic GW | Sampled GW | Spar-GW | NeuralGW |
|---|---|---|---|---|---|
| Time cost (s) | $5.7472 \pm 0.1189$ | $39.5246 \pm 3.6803$ | $58.6978 \pm 4.5848$ | $27.2126 \pm 1.1048$ | $2.6014 \pm 0.2182$ |

### D.4 SENSITIVITY ANALYSIS OF MMD LAYER

Here, we report the numerical performance of NeuralGW in approximating the GW distance when varing the number of MMDs in the MMD layer. It can be observed that the MMD layer is essential for enhancing the approximation capability of NeuralGW.

Table 9: Sensitivity on the number of MMDs in the MMD layer

| | MUTAG | BZR | COX2 | DHFR |
|---|---|---|---|---|
| NeuralGW: S=0 | $0.3087 \pm 0.0106$ | $0.3036 \pm 0.0157$ | $0.3108 \pm 0.0179$ | $0.2575 \pm 0.0075$ |
| NeuralGW: S=1 | $0.1578 \pm 0.0014$ | $0.2741 \pm 0.0789$ | $0.1544 \pm 0.0038$ | $0.2565 \pm 0.0145$ |
| NeuralGW: S=3 | $0.1251 \pm 0.0295$ | $0.1863 \pm 0.0608$ | $0.1432 \pm 0.0070$ | $0.1364 \pm 0.0274$ |
| NeuralGW: S=5 | $0.1458 \pm 0.0171$ | $0.1724 \pm 0.0688$ | $0.1506 \pm 0.0125$ | $0.1356 \pm 0.0266$ |
| NeuralGW: S=9 | $0.1414 \pm 0.0164$ | $0.1779 \pm 0.0445$ | $0.1466 \pm 0.0075$ | $0.1267 \pm 0.0077$ |

### D.5 ABLATION STUDY

As reviewed in Section 4, both GCN and Transformer have been used to approximate GW distances. NeuralGW is a hybrid GIN-Transformer neural network, which simultaneously combines the advantages of both. To support this motivation, we report the empirical results of ablation study on GIN/Transformer submodules. It can be found that this hybrid architecture would be more useful especially on complex datasets (*e.g.*, COX2 and DHFR) with the observation that It may not be the optimal choice for some simple datasets.

Table 10: Ablation study on GIN/Transformer submodules

| | MUTAG | BZR | COX2 | DHFR |
|---|---|---|---|---|
| Sampled GW | 0.4527 | 0.2916 | 0.4256 | 0.3223 |
| Spar-GW (8n) | 0.7677 | 1.0872 | 1.3896 | 1.0398 |
| Spar-GW (4n) | 0.6991 | 1.1227 | 1.5138 | 1.1161 |
| Spar-GW (2n) | 0.7982 | 1.1436 | 1.4421 | 1.1104 |
| NeuralGW$_{\text{GIN}}$ | $0.1454 \pm 0.0186$ | $0.2118 \pm 0.0857$ | $0.1514 \pm 0.0009$ | $0.2002 \pm 0.0395$ |
| NeuralGW$_{\text{MHA}}$ | $0.1281 \pm 0.0237$ | $0.1597 \pm 0.0397$ | $0.1487 \pm 0.0039$ | $0.1364 \pm 0.0152$ |
| NeuralGW | $0.1414 \pm 0.0164$ | $0.1779 \pm 0.0445$ | $0.1466 \pm 0.0075$ | $0.1267 \pm 0.0077$ |

### D.6 MORE RESULTS ON THE CROSS-DOMAIN PREDICTION SETTING

In Table 11, we provide the numerical results of the cross-domain prediction, where one dataset (rows) is for training and three datasets (columns) are for testing. We observe that the NeuralGW trained on other datasets performs better than the NeuralGW trained on MUTAG. One reason is that MUTAG (with 188 graphs) is much smaller than the other three datasets (each with more than 400 graphs).

Table 11: Cross-domain prediction from a single dataset (row/train) to another single dataset (column/test)

|  | MUTAG | BZR | COX2 | DHFR | AVG |
|---|---|---|---|---|---|
| trained on MUTAG | $0.1414 \pm 0.0164$ | $0.4569 \pm 0.0679$ | $0.9063 \pm 0.2501$ | $0.6575 \pm 0.0379$ | 0.5405 |
| trained on BZR | $0.3658 \pm 0.0475$ | $0.1779 \pm 0.0445$ | $0.1970 \pm 0.0360$ | $0.1832 \pm 0.0494$ | 0.2305 |
| trained on COX2 | $0.5691 \pm 0.0434$ | $0.3147 \pm 0.0111$ | $0.1466 \pm 0.0075$ | $0.2497 \pm 0.0280$ | 0.3200 |
| trained on DHFR | $0.5333 \pm 0.0251$ | $0.1866 \pm 0.0168$ | $0.1426 \pm 0.0027$ | $0.1267 \pm 0.0077$ | 0.2473 |

To evaluate the impact of diverse training data, we constructed a new set comprising molecular, bioinformatics, and social network datasets. Comparative results against a model trained solely on molecular data are presented in Table 12. The integrated dataset enhanced predictive performance in two out of three cases, demonstrating the utility of incorporating related domain data.

Table 12: Cross-domain prediction from multiple datasets (row/train) to a single dataset (column/test)

|  | DHFR | ENZYMES | IMDB |
|---|---|---|---|
| MUTAG | $0.6575 \pm 0.0379$ | $0.5985 \pm 0.0918$ | $0.8015 \pm 0.1703$ |
| MUTAG + DD + COLLAB | $0.6877 \pm 0.1247$ | $0.2301 \pm 0.1743$ | $0.3428 \pm 0.1672$ |

### D.7 MORE RESULTS AND VISUALIZATION ON SBM GRAPHS

We employed a Stochastic Block Model (SBM) to generate 200 graphs belonging to 5 classes, with 40 graphs per class. All graphs in 5 communities, while 5 classes differ in their underlying community structures. For illustration, we include five representative graphs (one from each class) to highlight the structural differences across classes (see Figure 3).

These 200 graphs were then split into training, validation, and test sets for the in-domain prediction task using NeuralGW. The full pairwise distance matrix, as well as the training, validation, test submatrices, are provided in Figure 4. One can see that the distributions of distances in the training, validation, and test sets are generally consistent with that of the full distance matrix.



(a) Class 1     (b) Class 2     (c) Class 3     (d) Class 4     (e) Class 5

Figure 3: Visualization of adjacency matrix of Sample graphs in 5 classes. (a) The adjacency matrix of a graph in Class 1; (b) The adjacency matrix of a graph in Class 2; (c) The adjacency matrix of a graph in Class 3; (d) The adjacency matrix of a graph in Class 4; (e) The adjacency matrix of a graph in Class 5

- We train NeuralGW on the training set and plot the averaged predicted GW distance matriix on the test set. As shown in Figure 5, the learned distances exhibit clear clustering patterns among graphs sharing the same community structure. This indicates that the learned GW distance successfully preserves meaningful structural information.
- We also provide a scatter plot comparing POT GW distances with NeuralGW-predicted distances on held-out test pairs (rightmost plot in Figure 5). The correlation coefficient is

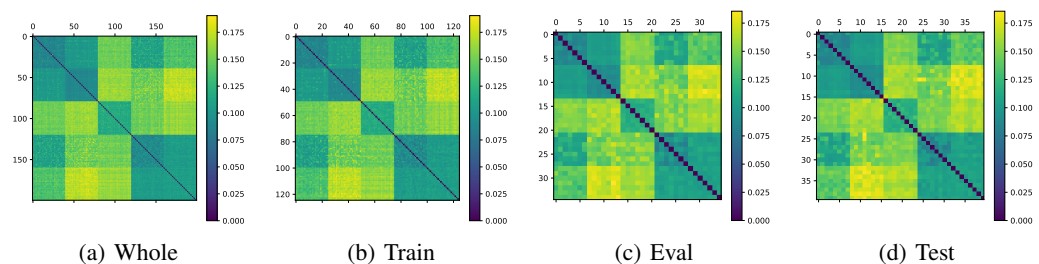

(a) Whole      (b) Train      (c) Eval      (d) Test

Figure 4: Matrix visualization of Groundtruth GW distances. (a) The whole GW distance matrix of $200 \times 200$; (b) The GW distance matrix of $125 \times 125$ for training; (c) The GW distance matrix of $35 \times 35$ for validation;(d) The GW distance matrix of $40 \times 40$ for testing

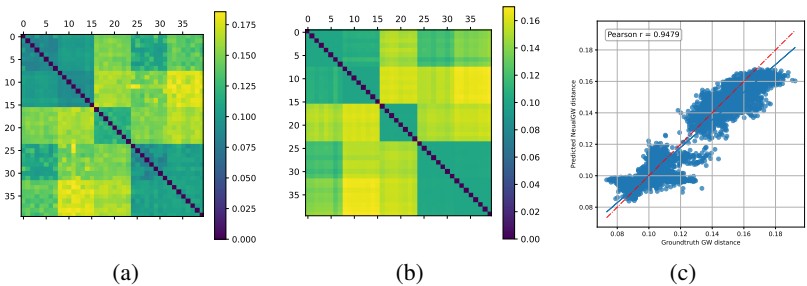

(a)      (b)      (c)

Figure 5: Comparison of NeuralGW prediction to the groundtruth GW distance. (a) Matrix visualization of Groundtruth GW matrix on held-out test pairs; (b) Matrix visualization of NeuralGW prediction on held-out test pairs; (c) Scatter plot comparing POT GW distance versus NeuralGW distance on held-out test pairs

close to 0.95, demonstrating that the predicted distances align well with the ground truth beyond the average error.

## D.8 INDOMAIN PREDICTION FOR LARGE GRAPHS

As shown in Figure 6, both the node distributions in COLLAB and REDDIT-BINARY are highly imbalanced. To evaluate the indomain prediction capability of NeuralGW on large graphs, we select graphs with $300 \sim 350$ nodes from each datasets. We report the performance of NeuralGW and NeuralGW-Naive trained on COLLAB$_{300\sim350}$ and REDDIT-BINARY$_{300\sim350}$, respectively. From Table 13, our proposed NeuralGW consistently outperforms the other two methods, demonstrating that the architecture of NeuralGW is both necessary and effective.

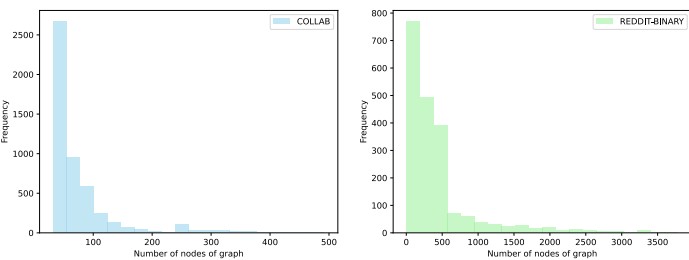

Figure 6: Histograms of two large datasets in TUDataset

Table 13: Average relative error of in-domain prediction

|  | COLLAB$_{300\sim350}$ | REDDIT-BINARY$_{300\sim350}$ |
|---|---|---|
| Spar-GW (8n) | 0.7676 | 16.4743 |
| Spar-GW (4n) | 0.7990 | 20.5937 |
| Spar-GW (2n) | 0.7727 | 21.8361 |
| NeuralGW-Naive | 1.1473 ± 0.9739 | 0.2576 ± 0.0167 |
| NeuralGW | 0.1748 ± 0.0116 | 0.1329 ± 0.0073 |

### D.9 CROSS-DOMAIN PREDICTION FOR LARGE GRAPHS

**From small graphs to large graphs** We report the performance of NeuralGW trained on combinations of datasets consisting of relatively small graphs and evaluated on datasets containing larger graphs. As shown in Figure 7, all graphs in those selected small datasets contain fewer than 80 nodes. Based on this observation, we train both NeuralGW and NeuralGW-Naive on these small datasets and evaluate them on larger graphs (*i.e.*, graphs with a greater number of nodes). In Table 14, we train both models on the combination of BZR+COX2+DHFR and test their performance on other graphs with more nodes each. For example, ENZYMES$_{\geq 70}$ refers to the subset of graphs from the ENZYMES dataset that contain more than 70 nodes.

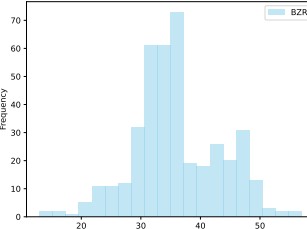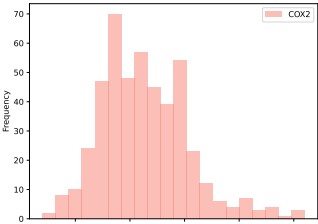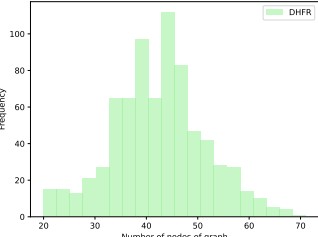

Figure 7: Histograms of three small datasets in TUDataset

Table 14: Cross-domain prediction for large graphs in bioinformatics

| Trainsets
Testset | BZR+COX2+DHFR
DD$_{300\sim350}$ | BZR+COX2+DHFR
ENZYMES$_{\geq 70}$ | BZR+COX2+DHFR
COLLAB$_{300\sim350}$ |
|---|---|---|---|
| Spar-GW (8n) | 1.1315 | 1.1438 | 0.7676 |
| Spar-GW (4n) | 1.2221 | 1.1945 | 0.7990 |
| Spar-GW (2n) | 1.0022 | 1.2004 | 0.7727 |
| NeuralGW-Naive | 0.5211 ± 0.0578 | 0.4133 ± 0.0472 | 816.2166 ± 183.8891 |
| NeuralGW | 2.2559 ± 0.2823 | 0.8766 ± 0.1540 | 0.6620 ± 0.0544 |

As shown in Table 14 the proposed NeuralGW generalizes well to ENZYMES$_{\geq 70}$ and achieves reasonable performance on DD$_{300\sim350}$. This behavior may be attributed to the fact that combination of chemical molecule datasets share structral and distributional similarities with ENZYMES, which also belongs to the bioinformatics domain. Furthermore, on COLLAB$_{300\sim350}$, NeuralGW generalizes effectively to this social network dataset despite being trained on chemical molecule datasets, and it exhibits greater stability than NeuralGW-Naive. Under the same configuration as NeuralGW except for the architecture difference, NeuralGW-Naive essentially fails to learning meaningful representations.

**From large graphs to large graphs** We here report the performance of NeuralGW trained on a dataset comprising relatively large graphs and evaluated on a separate dataset containing graphs of

comparable size. As in Table 15, we trained NeuralGW and NeuralGW-Naive on COLLAB$_{300\sim350}$ of 65 graphs and test it on REDDIT-BINARY$_{300\sim350}$ of 147 graphs. It shows a little difficult for NeuralGW to predict GW distance from one to another. One reason may be the significant difference between their sparsity of edges in each graph. Taking $1 - \frac{2\text{Avg. Edges}}{\text{Avg. Nodes}(\text{Avg. Nodes}-1)}$ as the indicator metric, it can be found from Table 5 that the average sparsity of edges in REDDIT-BINARY (0.9946) is much higher than that in COLLAB (0.1021). Note that our proposed NeuralGW takes as input the adjacency matrix and the corresponding embedding, which implies that the performance of NeuralGW can heavily rely on the sparsity of edges in the adjacency matrix. Consequently, it is very challenging for COLLAB and REDDIT-BINARY to do cross-domain prediction of GW distances.

Table 15: Cross-domain prediction between COLLAB$_{300\sim350}$ and REDDIT-BINARY$_{300\sim350}$

| Trainset
Testset | COLLAB$_{300\sim350}$
REDDIT-BINARY$_{300\sim350}$ | REDDIT-BINARY$_{300\sim350}$
COLLAB$_{300\sim350}$ |
|---|---|---|
| Spar-GW (8n) | 16.4743 | 0.7676 |
| Spar-GW (4n) | 20.5937 | 0.7990 |
| Spar-GW (2n) | 21.8361 | 0.7727 |
| NeuralGW-Naive | $20.4976 \pm 2.1248$ | $150.3923 \pm 21.0236$ |
| NeuralGW | $6.6194 \pm 1.9688$ | $0.9597 \pm 0.1494$ |

# E  EXTENSION: TRANSPORT PLAN PREDICTION

For the $S$ MMDs, we have $S$ kernel matrices respectively, denoted as $\mathcal{K}_{ij} := [\boldsymbol{K}_{ij}^{(1)}, \boldsymbol{K}_{ij}^{(2)}, \ldots, \boldsymbol{K}_{ij}^{(S)}] \in \mathbb{R}^{n_i \times n_j \times S}$. Let $\boldsymbol{P}_{ij} \in \mathbb{R}^{n_i \times n_j}$ be the transport plan between $G_i$ and $G_j$. We proposed to predict $\boldsymbol{P}_{ij}$ via utilizing $\mathcal{K}_{ij}$. The model is formulated as

$$\hat{\boldsymbol{P}}_{ij} = \xi_\varphi(\mathcal{K}_{ij}) \tag{16}$$

where $\xi : \mathbb{R}^S \to \mathbb{R}$ is an MLP with parameters $\phi$ applied to the third dimension of $\mathcal{K}_{ij}$.

To be more specific, as in Figure 8, after getting the embeddings $\boldsymbol{Z}_i$ and $\boldsymbol{Z}_j$ of nodes of $G_i$ and $G_j$, we first construct a sequence of kernels $\{\boldsymbol{K}_s(\boldsymbol{Z}_i, \boldsymbol{Z}_j)\}_{s=1}^S$, which are then vectorized and concatenated to be the latent representations of each entry in the transport plan. This derived representation is fed into the MLP for predicting the final transport plan. Then we solve the following problem:

$$\underset{\theta,\varphi}{\text{minimize}} \ \frac{2}{N(N-1)} \sum_{i=1}^N \sum_{j=i+1}^N \ell\left(\hat{\boldsymbol{P}}_{ij}, \boldsymbol{P}_{ij}\right) \tag{17}$$

where $\ell$ denotes a loss function and $\hat{\boldsymbol{P}}_{ij}[k,l]$ denotes the element $(k,l)$ of matrix $\hat{\boldsymbol{P}}_{ij}$. We call it NeuralGW-TP. Note that we can predict the transport plan and GW distance jointly or separately. One can use the norm of the deviance or the KL divergence to be as a loss function:

$$\ell\left(\hat{\boldsymbol{P}}_{ij}, \boldsymbol{P}_{ij}\right) = \frac{\|\hat{\boldsymbol{P}}_{ij} - \boldsymbol{P}_{ij}\|}{\|\boldsymbol{P}_{ij}\|} \ \text{or KLD}\left(\hat{\boldsymbol{P}}_{ij}, \boldsymbol{P}_{ij}\right) \tag{18}$$

**Relative error of approximation on transport plan** We report the performance of NeuralGW-TP in terms of their relative error of approximation on transport plan matrix. That is, given the ground-truth Gromov-Wasserstein transport plan $\boldsymbol{P}$, the relative residual error of an approximation transport plan $\boldsymbol{P}_{\text{Pred}}$ is defined as

$$\mathcal{E}_{\boldsymbol{P}_{GW}}(\boldsymbol{P}_{\text{Pred}}) = \frac{\|\boldsymbol{P}_{\text{Pred}} - \boldsymbol{P}_{\text{GW}}\|_1}{\|\boldsymbol{P}_{\text{GW}}\|_1}$$

for which $\boldsymbol{P}_{\text{GW}}$ can be based on the output of POT. A lower $\mathcal{E}_{\boldsymbol{P}_{GW}}$ indicates a better performance in predicting the transport plan of GW.

To empirically show the validity of our proposed NeuralGW-TP in Section **??**, we here report the relative error of approximate on transport plans of graph datasets. As shown in Table 16, our NeuralGW-TP can provide better or comparable approximation of transport plans to Spar-GW. In this context, the time cost for NeuralGW-TP refers to its training period.

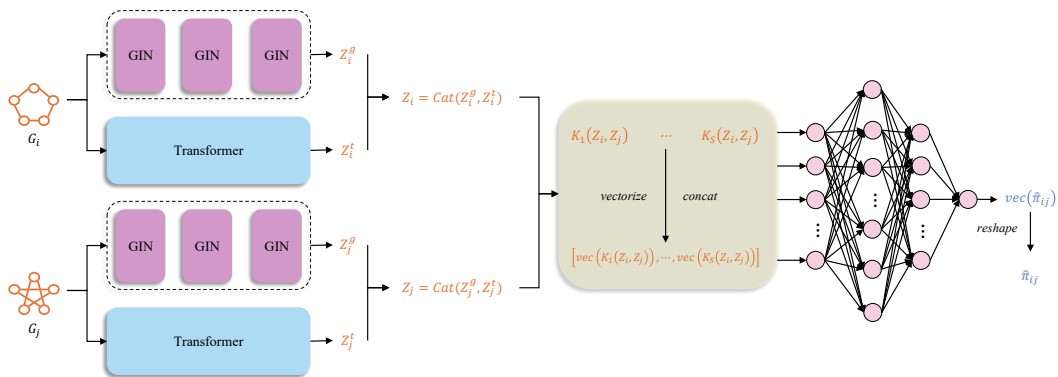

Figure 8: Architecture of NeuralGW-TP. On the left, the GINs and Transformers are shared by the two input graphs.

Table 16: Indomain prediction for transport plan

| | MUTAG | | BZR | |
|---|---|---|---|---|
| | Error | Time(m) | Error | Time(m) |
| Spar-GW (8n) | 1.3754 | $\sim 28$ | 1.4375 | $> 60$ |
| Spar-GW (4n) | 1.3671 | $\sim 18$ | 1.4140 | $> 60$ |
| Spar-GW (2n) | 1.3754 | $\sim 6$ | 1.4220 | $\sim 27$ |
| NeuralGW-TP | $1.3111 \pm 0.1398$ | $< 5$ | $1.0921 \pm 0.0450$ | $< 5$ |

