# OpenReview forum: "Bridging Graph Worlds: Neural Approximation of Gromov-Wasserstein Distances"
_ICLR.cc/2026/Conference — Submitted to ICLR 2026_

### Official Review · Reviewer_ioen · 2025-10-20

**Soundness:** 2
**Presentation:** 1
**Contribution:** 3
**Rating:** 4
**Confidence:** 3

**Summary:**

Assuming that a collection of N graphs is given, as well as the pairwise "true" Gromov-Wasserstein (GW) distance between them, the main contribution of this paper is the introduction of a neural architecture combining GINs and Transformer layers in order to i) learn the GW distance between the graph training pairs (with the aim to generalize to new unseen test pairs) ii) learn the optimal transport plans between graph pairs. Although the idea is not entirely original, as the author acknowledge (NeuralGWD-Naive), their architecture outperform previous alternatives via the introduction of an original Maximum Mean Discrepancy (MMD) layer. A theoretical result, bounding the risk linked to the proposed estimator is claimed and proved and several experiments highlight the improved performance of the proposed approach together with its reduced computation cost, notably with respect to a direct calculation of the GW distance.

**Strengths:**

Without no doubt, the line of research proposed by the authors is of interest, since a direct computation of the optimal transports distances, in general, is quite prohibitive.

**Weaknesses:**

The main weakness of the paper is that it is really written too quickly and/or with some errors making it impossible to assess whether the main claims are entirely true or not.

**Questions:**

Main points:

i) At line 268 and following authors claim that NeuralGW induces a pseudo-metric. There is a proof for it? For instance, point 1. requires that your architecture is permutation invariant/equivariant. Is it the case?
Moreover point 4 (l. 299) is meaningless: you state $d_{NGW}(G_1, G_2) = d_p(G_1, G_2)$. However you stated a few lines before that $(\mathcal{G}_p, d_p)$ is a metric space, so $d_p$ measures the distance between the nodes of $\mathcal{G}_p$ and *not* the distance between two graphs;

ii) I take Th. 1 as given (I'll be back to the proof in a while). Let me call $\mu$ the upper bound on the right hand side of Eq. (10). At line 325 you state  "We have $U_N(\ell_{\hat{f}}) - R(\ell_{f^*})  \leq 2\mu$".  Now, my guess is:

$ U_N(\ell_{\hat{f}}) - R(\ell_{f^*}) = (U_N(\ell_{\hat{f}}) - U_N(\ell_{f^*)) + (U_N(\ell_{f^*) -  R(\ell_{f^*})) $

the first term on the r.h.s. of the equality is negative by definition of $ f^* $ and hence the quantity on the l.h.s. is majored by  $ U_N(\ell_{f^{\*}}) -  R(\ell_{f^{\*}}) \leq \mu $.

So where does the 2 in front of $\mu$ come from?

iii) At line 337 you state: "the time complexity in the optimization is $\mathcal{O}(Bn^2dL + \dots )$". Where does this term come from?? Even assuming that your architecture simply is a MLP with width $\overline{d} \geq d $ and depth $L$: your positional encodings for one graph form a matrix $n \times d$ to be processes by a matrix $d \times \overline{d}$ in the first layer and $\overline{d} \times \overline{d}$ later on, for $L$ times, which gives $\mathcal{O}(n\overline{d}L)$. You simply consider that $n \leq n^2$? Moreover your architectures are not simple MLPs? And the gradient calculations? Please detail.

iv) In the experiments: unless I am wrong, you never specify the size of your trining data set, which is crucial since you need to compute $N_{train}^2$ GW distances with POT in order to have your ground truth...

v) About appendix A. I tried to read it. Ok, I acknowledge that I am not someone who bounds empirical risks everyday, so my expertise might be too limited in order to fully understand this section. However... At line 618: there is something missing on the r.h.s. of the inequality? Maybe $\sum_{I=1}^{N/2}$? In the proof of that lemma you care about the first term $U_N(\ell_f)$. What about the second? At line 629 you meant "to move sup *inside* the permutation average? Moreover: what is $\mathcal{R}_N(\mathcal{F})$ at line 634 ? It is never defined.

vi) About appendix C. I think in Eq. (13) the second sum should be $\sum_{j = i+1}^N$ and, more important, is your estimated transport plan admissible? How to be sure that the marginals are fulfilled?

Minor remarks:

i) Be careful around line 041: when adding an entropic regularization term the Wasserstein distance is no longer a distance, so the term "sinkhorn distance" might be misleading.
ii) line 106: it should be $GW^2(\mu, \nu)$ in place of $GW^2(X, Y)$.
iii) line 158: it should be $\alpha_{uv}$ in place of $\alpha_{ij}$
iv) Eq. (9): who is $\theta$?

---

> ### Author Response · Authors · 2025-11-28
>
> The authors thank the reviewer for the comments. Our responses to your comments and questions are as follows.
>
> __Response to Weakness 1__:
>
> Thank you for pointing out these issues. We clarify the permutation-related properties of our model as follows.
>
> NeuralGW is permutation invariant by design. This follows from two key observations:
>
> * Under the graph learning setting, GIN and Transformer layers are permutation equivariant, meaning that permuting the input node order results in an identical permutation of the output node representations;
> * MMD is permutation invariant, as it depends only on pairwise similarities between sets and is therefore unaffected by the ordering of elements
>
> Since the MMD module serves as a set-level readout layer, it produces graph-level representations that are themselves permutation invariant. The subsequent MLP operates on these graph-level embeddings and, being applied after an invariant readout, cannot break permutation invariance. Therefore, the overall NeuralGW architecture is permutation invariant.
>
> Regarding Property 4, we are sorry for any confusion. It has been removed from Proposition 1.
>
> We thank the reviewer again for these valuable comments and have added the whole proof in the appendix.
>
> __Response to Weakness 2__:
>
> Thank you for pointing out this typo. It should have been
> $$
> R(\ell _{\hat{f}}) - R(\ell _{f^*}) \le 2\sup _{\ell _f\in\mathcal{L}}|R(\ell _f) - U _N(\ell _f)|
> $$
>
> which is the standard uniform-deviation argument. To make it clearer, we present the proof on this inequality and associate it with the bound on $U _N(\ell _{\hat{f}}) - R(\ell _{f^*})$.
>
> Fix notation and define the uniform deviation $\mu\coloneq \sup_{\ell_f\in\mathcal{L}}|R(\ell_f) - U_N(\ell_f)|$. So for every $f$, we have
> \begin{align*}
> -\mu\le R(\ell_f) - U_N(\ell_f) \le \mu \iff U_N(\ell_f) - \mu \le R(\ell_f) \le U_N(\ell_f) + \mu
> \end{align*}
>
> Let $\hat{f}=\mathop{\arg\min} _f U _N(\ell _f)$ be the minimizer of the data-dependent criterion $U _N$ and let $f^*=\mathop{\arg\min} _f R(\ell _f)$ be the minimizer of population risk. We start from the following decomposition.
>
> $$
> \begin{aligned}
> & R(\hat{f}) - R(f^*) \\\\
> = & (R(\hat{f}) - U _N(\hat{f})) + (U _N(\hat{f}) - U _N(f^ *)) + (U _N(f^ *) - R(f^ *))\\\\
> \le & \mu + 0 + \mu = 2\mu
> \end{aligned}
> $$
>
> where the first term $R(\hat{f}) - U _N(\hat{f}) \le \mu$ because $|R(\ell _{\hat{f}}) - U _N(\ell _{\hat{f}})|\le\mu$; the second term $U _N(\hat{f}) - U _N(f^ *) \le 0$ bcause $\hat{f}$ minimizes $U _N$; the third term $U _N(f^ *) - R(f^ *)\le\mu$ again by the uniform bound.
>
> Therefore, we proved the usual empirical risk minimization (ERM) guarantee:
> $$
> R(\ell _{\hat{f}}) - R(\ell _{f^*}) \le 2\sup _{\ell_f\in\mathcal{L}}|R(\ell _f) - U_N(\ell _f)|
> $$
>
> Next, we relate $U _N(\ell _{\hat{f}}) - R(\ell _{f^*})$ to this guarantee as follows.
>
> * Decompose it by inserting $R(\ell_{\hat{f}})$:
> $$
> U_N(\ell _{\hat{f}}) - R(\ell _{f^ *}) = (U _N(\ell _{\hat{f}}) - R(\ell _{\hat{f}})) + (R(\ell _{\hat{f}}) - R(\ell _{f^ *})) \le \mu + 2\mu = 3\mu
> $$
> * Decompose it by inserting $U _N(\ell _{f^ *})$:
> $$
> U _N(\ell _{\hat{f}}) - R(\ell _{f^ *}) = (U _N(\ell _{\hat{f}}) - U _N(\ell _{f^ *})) + (U _N(\ell _{f^ *}) - R(\ell _{f^ *})) \le 0 + \mu = \mu
> $$
>
> Thus, the tightest bound is $U_N(\ell_{\hat{f}}) - R(\ell_{f^*}) \le \mu$.
>
> In summary, one should have
> $$
> \begin{aligned}
> & R(\ell _{\hat{f}}) - R(\ell _{f^ *}) \le 8M\psi(1 + \log\frac{4M}{\psi}) + 8M^2\sqrt{8N^{-1}\log(2/\delta)}\\\\
> & U_N(\ell _{\hat{f}}) - R(\ell _{f^ *}) \le 4M\psi(1 + \log\frac{4M}{\psi}) + 4M^2\sqrt{8N^{-1}\log(2/\delta)}
> \end{aligned}
> $$
> Thank you again for pointing out this issue.

---

> ### Author Response · Authors · 2025-11-28
>
> __Response to Weakness 3__:
>
> We assumed that the width and depth of GIN, Transformer, and MLP are all $\bar{d}$ and $L$, respectively.
>
> * For MLP, to process one graph, the per-layer complexity is $\mathcal{O}(n\bar{d}^2)$.
> * For GIN, to process one graph, the per-layer complexity is $\mathcal{O}(n^2\bar{d}+n\bar{d}^2)$, due to the neighbor aggregation operation and feedforward operation.
> * For Transformer, to process one graph, the per-layer complexity is $\mathcal{O}(n\bar{d}^2+n^2\bar{d}+n^2\bar{d})$, where the three terms are due to weight matrix multiplications, the construction of the attention map, and the multiplication with the attention map, respectively.
> * The complexity of gradient calculations is about two times that of the forward pass.
>
> Summing up these terms and multiplying with $L$, we have $\mathcal{O}((n\bar{d}^2+n^2\bar{d})L)$ for processing one graph in one iteration. By assuming $\bar{d}\leq n$, the complexity is simplified to $\mathcal{O}(n^2\bar{d}L)$. For $N$ graphs and $T$ iterations, as well as the complexity for all position encoders and the MMD layer, we have a complexity of $\mathcal{O}\left(N n^2 \bar{d}+\left(N n^2 \bar{d} L+N^2 S n^2 \bar{d}\right) T\right)$. We added these details to the appendix.
>
> __Response to Weakness 4__:
>
> We are sorry for this missing information. We have summarized the basic details of the datasets in Table 5 of Appendix B.1.
>
> For in-domain prediction, each dataset is split into training, validation, and test sets with proportions 64:16:20. For cross-domain prediction, the training set consists of all graphs from the source datasets while the validation and test sets are obtained by splitting the target set in a 20:80 ratio.
>
> Neural networks can easily memorize the training data, especially when they have high capacity. In our cross-domain prediction setting, NeuralGW tends to fit the source datasets perfectly, which may harm its ability to generalize to the target dataset. A validation set is therefore necessary to monitor and prevent this degradation in cross-domain generalization.
>
> If we used a validation set taken from a small subset of the source dataset, early stopping would only protect generalization within the source domain. But we have no guarantee that this reflects generalization to the target domain, whose distribution is different and whose deviation from the source is unknown.
>
> To properly track cross-domain generalization, we select a validation subset from the target dataset itself and evaluate NeuralGW's performance on this subset during training. When this validation performance stops improving, we stop training. Since both the validation set and the test set come from the same target distribution, improving performance on the validation set is a reliable indicator of improving performance on the test set.
>
> For this reason, the validation set were drawn from the target dataset, not the source dataset.
>
> We have added these detail to Appendix B.1 for clarity.

---

> > ### Author Response · Authors · 2025-11-28
> >
> > __Response to Weakness 5__:
> >
> > We are sorry for the seemingly lengthy proof. To make it clearer, we present the detailed proof of Lemma 1 below.
> >
> > **Proof of Lemma 1**
> >
> > __Setup and notation__ Let entries of $X = (X_1, X_2, \dots, X_N)$ be _i.i.d._ random variables taking values in some space $\mathcal{X}$. Let $\ell_f:\mathcal{X}\times\mathcal{X}\to\mathbb R$ be a symmetric kernel (_i.e._, $\ell_f(x, y) = \ell_f(y,x)$). The U-statistic of order 2 is
> > $$U_N(\ell_f) = \frac{2}{N(N-1)}\sum_{1\le i<j \le N}\ell_f(X_i, X_j).$$
> > Denote $m = \lfloor N/2\rfloor$ (For simplicity, assume $N$ is even so $m=n/2$; nothing essential will change otherwise.)
> >
> > We want to relate $U_N(\ell_f)$ to averages of independent pair sums (or to Rademacher symmetrized sums) via a combinatorial permutation argument.
> >
> > 1) __The permutation/pairing representation__
> >
> > Consider the set $\mathfrak{S} _n$ of all permutations $\pi$ of $[n]$. For each permutation $\pi$, define the paired-sum statistic
> >
> > $$S_\pi(\ell_f) = \frac{1}{m}\sum_{k=1}^m\ell_f(X_{\pi(2k - 1)}, X_{\pi(2k)}).$$
> >
> > Then, Hoeffding, 1963 claims that
> > $$
> > U _N(\ell _f) = \frac{1}{|\mathfrak{S} _N|}\sum _{\pi\in\mathfrak{S} _N} S _\pi(\ell _f)
> > $$
> > _i.e._, the U-statistic is the average, over all permutations, of the paired-sum statistic.
> >
> > 2) __Symmetrization for a fixed permutation__
> >
> > Fix a permutation $\pi$. Consider the centered quantity $S_\pi(h) - \mathbb{E}[S_\pi(\ell_f)]$. We can symmetrize this difference using an independent sample $X' = (X_1', X_2', \dots, X_N')$ (_i.i.d._ copies of $X_i$) and Rademacher signs. A standard inequality is
> >
> > $$
> > \mathbb{E}[\sup _{\ell _f\in\mathcal{L}}|S _\pi(\ell _f) - \mathbb{E}S _\pi(\ell _f)|] \le \mathbb{E} _{X,X'}\mathbb{E} _\sigma[\sup _{\ell _f\in\mathcal{L}}|\frac{1}{m}\sum _{i=1}^m\sigma _k(\ell _f(X _{\pi(2k - 1)}, X _{\pi(2k)}) - \ell _f(X' _{\pi(2k - 1)}, X' _{\pi(2k)}))|]
> > $$
> >
> > where $\sigma = (\sigma_1,\dots,\sigma_m)$ are _i.i.d._ Rademacher signs, independent of $X,X'$.
> >
> > Since $|a - b| \le |a| + |b|$ and the two halves are identically distributed, one has
> >
> > $$
> > \mathbb{E}[\sup _{\ell _f\in\mathcal{L}}|S _\pi(\ell _f) - \mathbb{E}S _\pi(\ell _f)|] \le 2\mathbb{E} _X\mathbb{E} _\sigma[\sup _{\ell _f\in\mathcal{L}}|\frac{1}{m}\sum _{i=1}^m\sigma _k\ell _f(X _{\pi(2k - 1)}, X _{\pi(2k)})|]
> > $$
> >
> > So for each fixed $\pi$, the deviation of $S_\pi$ is controlled by a Rademacher average over the $m$ paired terms.
> >
> > 3) __Average over permutations to get the bound for U-statistic__
> >
> > Since $U_N(\ell_f)$ is the average of $S_\pi(\ell_f)$ over permutations, one has
> >
> > $$
> > U _N(\ell _f) - \mathbb{E}[U _N(\ell _f)] = \frac{1}{|\mathfrak{S} _N|}\sum _{\pi\in\mathfrak{S} _N}(S _\pi(\ell _f) - \mathbb{E}[S _\pi(\ell _f)])
> > $$
> >
> > By convexity of the sup and Jensen (or simply triangular inequality for expectation of sup and averages), we get
> >
> > $$
> > \mathbb{E}[\sup _{\ell_f\in\mathcal{L}}|U _N(\ell _f) - \mathbb{E}[U _N(\ell _f)]|] \le \frac{1}{|\mathfrak{S} _N|}\sum _{\pi\in\mathfrak{S} _N}\mathbb{E}[\sup _{\ell _f\in\mathcal{L}}|S _\pi(\ell _f) - \mathbb{E}[S _\pi(\ell _f)]|].\\
> > $$
> >
> > (**move sup inside the permutation average**)
> >
> > Combining with the bound for each $\pi$ from Part 2) yields
> >
> > $$
> > \begin{aligned}
> > \mathbb{E}[\sup _{\ell _f\in\mathcal{L}}|U _N(\ell _f) - \mathbb{E}[U _N(\ell _f)]|] & \le \frac{2}{|\mathfrak{S} _N|}\sum _{\pi\in\mathfrak{S} _N}\mathbb{E} _X\mathbb{E} _\sigma[\sup _{\ell _f\in\mathcal{L}}|\frac{1}{m}\sum _{i=1}^m\sigma _k\ell _f(X _{\pi(2k - 1)}, X _{\pi(2k)})|]\\\\
> > & = 2\mathbb{E} _X\mathbb{E} _\sigma[\sup _{\ell_f\in\mathcal{L}}|\frac{1}{m}\sum _{i=1}^m\sigma _k\ell _f(X _{2k - 1}, X _{2k})|]
> > \end{aligned}
> > $$
> >
> > where the last identity holds because the distribution of the paired terms $(X_{\pi(2k - 1)}, X_{\pi(2k)})$ is the same for every $\pi$ (_i.e._, permuting indices does not change joint distribution). In other words, the permutation average collapses to the expected Rademacher average over one arbitrary fixed pairing of the sample into $m$ disjoint pairs (because all pairings are exchangeable).
> >
> > Therefore, the final symmetrization/decoupling inequality for U-statistic of order 2 is typically stated as
> >
> > $$
> > \mathbb{E}[\sup _{\ell _f\in\mathcal{L}}|U _N(\ell _f) - \mathbb{E}[U _N(\ell _f)]|] \le 2\mathbb{E} _X\mathbb{E} _\sigma[\sup _{\ell _f\in\mathcal{L}}|\frac{1}{m}\sum _{i=1}^m\sigma _k\ell _f(X _{2k - 1}, X _{2k})|]
> > $$

---

> ### Author Response · Authors · 2025-11-28
>
> __Response to Weakness 5 (continued)__:
>
> In summary, we highlight our response to these questions below.
>
> * At line 618: there is something missing on the r.h.s. of the inequality? Maybe $\sum_{I=1}^{N/2}$?
>
> __Response__: Yes, indeed. We have updated the inequality.
>
> * In the proof of that lemma you care about the first term $U_N(\ell_f)$. What about the second?
>
> __Response__: The second term $R(\ell_f)$ is treated as $\mathbb{E}[U_N(\ell_f)]$ (true risk as given in the above proof).
>
> * At line 629 you meant "to move sup inside the permutation average?
>
> __Response__: We are sorry for this wrong description. You are right. We have corrected it.
>
> * Moreover: what is $\mathcal{R}_N(\mathcal{F})$ at line 634 ?
>
> __Response__: We are sorry for any confusion. $\mathfrak{R} _N(\mathcal{F}) = \mathbb{E} _\sigma[\sup _{f\in\mathcal{F}}\frac{1}{N}\sum _{i=1}^N\sigma _i f(X _i)]$ is the empirical Rademacher complexity of the function class $\mathcal{F}$ where $f\in\mathcal{F}$. We have added its definition in Lemma 2.
>
> We appreciate your careful examination of our proof.
>
> [1] Wassily Hoeffding. Probability inequalities for sums of bounded random variables. Journal of the
> American statistical association, 58(301):13–30, 1963
>
> __Response to Weakness 6__:
>
> Thanks for pointing out this typo. In our experiments, uniform marginals are adopted when preparing the ground-truth GW pairwise distances as supervised information. The squared loss encourages $\|\hat{\mathbf P} - \mathbf P\|_F \to 0$, which, by the properties of Frobenius norm, implies $\hat{\mathbf P} \to \mathbf P$. In other words, minimizing the squared loss enforces that the learned transport plan respects the uniform marginals.
>
> An alternative way to explicitly enforce uniform marginals on the predicted transport plan $\hat{\mathbf P}$ is via Sinkhorn-like alternating steps:
>
> * $\hat{\mathbf P} \leftarrow \frac{\hat{\mathbf P}}{\text{nrow}(\hat{\mathbf P})\hat{\mathbf P}\mathbf 1_{\text{ncol}(\hat{\mathbf P})}}$ which normalizes $\hat{\mathbf P}$ along the first dimension;
> * $\hat{\mathbf P} \leftarrow \frac{\hat{\mathbf P}}{\text{ncol}(\hat{\mathbf P})\hat{\mathbf P}\mathbf 1_{\text{nrow}(\hat{\mathbf P})}}$ which normalizes $\hat{\mathbf P}$ along the second dimension;
>
> These operations follow standard Python broadcasting rules. By the convergence properties of Sinkhorn iterations, $\hat{\mathbf P}$ will converge to a transport plan with uniform marginals.
>
> __Response to Minor remarks__:
>
> Thanks for your careful review. We have corrected them.

---

### Official Review · Reviewer_qoyc · 2025-10-28

**Soundness:** 3
**Presentation:** 3
**Contribution:** 2
**Rating:** 4
**Confidence:** 2

**Summary:**

The paper studies the Gromov-Wasserstein Distances for comparing two graphs and proposes an approximation using a neural network. The proposed architecture combines the well-known GIN (graph isomorphism network) and a transformer module. The embedded graphs are then compared as empirical distributions using multiple maximum mean discrepancy (MMD) values.

**Strengths:**

The paper addresses an important topic, which is the estimation of graph distances. It explores recent tools, such as graph isomorphism networks and transformers.

**Weaknesses:**

The choice of using the graph isomorphism network (GIN) is motivated by its expressive power. However, they are comparable to the Weisfeiler–Lehman (WL) graph isomorphism test, namely they are as powerful as the 1-WL test in graph discrimination. This might not be enough in general, as one would require more discriminative power, beyond the 1-WL test.

It is well-known that GIN, as well as other related graph neural networks, suffer from over-smoothing issues. It is not clear how the proposed method overcomes this issue.
While it would be recommended to have fewer layers, the impact of the number of layers needs to be better analyzed in the light of Theorem 1 (namely the increase of $K_1$).

Considering the MLP at the end of the architecture (not the one of the GIN), it is written in the main body that the MLP has only one layer at Page 6. Figure 1 shows 2 layers. Experimental settings as given in Page 15 consider a 3-layer MLP.

In the analysis of the computational complexity, it is written “If we treat L and S as constants”. It is not clear what the authors mean. Technically, they are not variables, but they are hyperparameters to be set by the user.

The provided numerical results are missing comparative analysis with related methods. The proposed method has essentially a Siamese architecture, which would be similar to many other methods from the literature, including those cited in Section 4.2 (and many others). The authors have chosen to compare to only the Wasserstein Wormhole. The other compared methods are sampled GW and Spar-GW, which are less related to the current architecture and are a bit old.

There are some sentences that are poorly written, such as “Modeling a graph … is a choice for comparing graph pairs”, “The aforementioned methods are lacking of …”…

**Questions:**

No further questions.

---

> ### Author Response · Authors · 2025-11-28
>
> The authors thank the reviewer for the comments. Our responses to your comments and questions are as follows.
>
> __Response to Weakness 1__ (**1-WL test**):
>
> We appreciate your insightful comment. Some studies [1-3] have proved that well-designed GNNs and graph transformers are as powerful as the $k$-WL test, where $k>1$. Therefore, we could consider these more expressive models. The reason we didn't use them is two-fold.
>    * First, most $k$-WL GNNs have much higher computational complexity than those used in our paper, but we focus on fast approximation for the GW distance.
>    * Second, in most datasets, the graphs have different numbers of nodes and hence can be easily distinguished by the 1-WL test. We aim to construct a continuous and informative distance measure between two graphs, rather than a hard zero-one judgment. The GIN and transformer we used in our paper have shown effective performance in many graph learning tasks.
> Nevertheless, as you suggested, we will continuously try to use more effective GNN models to improve the prediction accuracy for the GW distance.
>
> [1] https://arxiv.org/pdf/2207.02505
>
> [2] https://arxiv.org/pdf/2301.09505
>
> [3] https://arxiv.org/pdf/2406.03148
>
> __Response to Weakness 2__ (**over-smoothing**):
>
>
>
> Thanks for your thoughtful suggestion. Our NeuralGW model adopts a hybrid GIN-Transformer architecture, where each GIN layer is equipped with a residual (skip) connection, and the GIN and Transformer branches operate in parallel. For clarity, when using one GIN layer and one Transformer layer, and the forward computation before entering the MMD layer is:
> $$
> \begin{aligned}
> \mathbf{Z}_1 & \leftarrow \mathrm{GINConv}(\mathbf X, \mathrm{edge\\_index}) + \mathbf{X} \\\\
> \mathbf{Z}_2 & \leftarrow \mathrm{TransformerEncoderLayer}(\mathbf X)\\\\
> \mathbf{Z} & \leftarrow [\mathbf Z_1,\mathbf Z_2]
> \end{aligned}
> $$
>
> Both the skip connections and the parallel combination of GIN and Transformer paths help alleviate the over-smoothing phenomenon typically observed in deep GNNs.
>
> We provide two sets of numerical results to support this:
> * __Ablation study__ Using one GIN layer and one Transformer layer as a representative case, we conduct an ablation study on these two components. As shown in Table 1, the Transformer consistently improves the performance of the GIN, indicating that the Transformer effectively complements the local structural learning of the GIN.
> * __Sensitivity analysis__: We further investigate the effect of stacking multiple GIN layers while fixing the number of Transformer layers to 1. Table 2 shows that NeuralGW maintains stable performance across different numbers of GIN layers, suggesting robustness against the over-smoothing issues that typically arise when deepening GIN-based architectures.
>
> Taken together, these results demonstrate that the NeuralGW architecture effectively mitigates over-smoothing and remains stable even with multiple GIN layers.
>
> __Table 1__: Ablation study on GIN/Transformer submodules
> |                                           | MUTAG             | BZR               | COX2              | DHFR              |
> |-------------------------------------------|-------------------|-------------------|-------------------|-------------------|
> | Sampled GW                                | 0.4527            | 0.2916            | 0.4256            | 0.3223            |
> | Spar-GW (8n)                              | 0.7677            | 1.0872            | 1.3896            | 1.0398            |
> | Spar-GW (4n)                              | 0.6991            | 1.1227            | 1.5138            | 1.1161            |
> | Spar-GW (2n)                              | 0.7982            | 1.1436            | 1.4421            | 1.1104            |
> | $\text{NeuralGW}_\text{GIN-only}$         | $0.1454\pm0.0186$ | $0.2118\pm0.0857$ | $0.1514\pm0.0009$ | $0.2002\pm0.0395$ |
> | $\text{NeuralGW}_\text{Transformer-only}$ | $0.1281\pm0.0237$ | $0.1597\pm0.0397$ | $0.1487\pm0.0039$ | $0.1364\pm0.0152$ |
> | NeuralGW                                  | $0.1414\pm0.0164$ | $0.1779\pm0.0445$ | $0.1466\pm0.0075$ | $0.1267\pm0.0077$ |
>
> __Table 2__: Impact of the number of GIN layers
> |       | MUTAG             | BZR               | COX2              | DHFR              |
> |-------|-------------------|-------------------|-------------------|-------------------|
> | GIN-1 | $0.1414\pm0.0164$ | $0.1779\pm0.0445$ | $0.1466\pm0.0075$ | $0.1267\pm0.0077$ |
> | GIN-3 | $0.2149\pm0.0594$ | $0.1435\pm0.0364$ | $0.1298\pm0.0073$ | $0.1140\pm0.0016$ |
> | GIN-5 | $0.1737\pm0.0503$ | $0.1368\pm0.0344$ | $0.1343\pm0.0070$ | $0.1699\pm0.0718$ |
> | GIN-7 | $0.1804\pm0.0534$ | $0.1425\pm0.0282$ | $0.1329\pm0.0041$ | $0.1127\pm0.0034$ |
> | GIN-9 | $0.1758\pm0.0366$ | $0.1598\pm0.0426$ | $0.1275\pm0.0053$ | $0.1183\pm0.0093$ |

---

> ### Author Response · Authors · 2025-11-28
>
> __Response to Weakness 3__ (**MLP layers**):
>
> We are very sorry for the inconsistent description in the manuscript. The final MLP module can indeed adopt a flexible structure depending on the specific dataset. In all of our experiments, we used a 3-layer MLP. On Page 6, the $L_{\text{MLP}}$ in the theorem means the Lipschitz constant of the entire MLP, not a single layer. We have revised the corresponding descriptions throughout the paper and updated Figure 1 to ensure consistency and clarity.
>
> __Response to Weakness 4__ (**time complexity**):
>
>
>
> We'd like to clarify that the complexities of the baselines (e.g., GW) shown in the table are per-iteration complexity, where the number of iterations, denoted as $I$, hasn't been included. To make the comparison fair, we treat $L$ and $S$ of our NeuralGW as constants, and they are often less than $I$. We added an explanation to the section (Section 3.4).
>
> __Response to Weakness 5__ (**related methods**):
>
>
>
> Our work focuses on the natural approximation of GW distances between graphs. It is therefore natural to compare the proposed NeuralGW with both neural network-based approaches and traditional optimization-based methods.
>
> A few recent works are studying or accelerating the entropic GW, which is an approximation of GW but often leads to a much larger GW distance. For reference, we provide a numerical comparison between the entropic GW and the standard GW in our response to W4 of Reviewer u6um. The entropic GW distance can become significantly larger than the POT-based GW distance when the regularization parameter stays far away from zero.
>
> Our work focuses specifically on accelerating the original GW without entropic regularization. We have tried our best to find more baselines or competitors for our NeuralGW. Sampled GW and Spar-GW, as discussed in [1], approximate the original GW distance via distributional sub-sampling. Therefore, we included them as classical optimization baselines.
> Regarding the neural network-based methods, DWE [2] and Wormhole [3] are the most conceptually related to our approach. However, DWE is implemented as a CNN-based Siamese network and therefore applies only to 2D/3D images or point clouds with fixed-grid support, which limits its applicability to general graphs. For this reason, we did not include DWE in our benchmark. We agree that some other model architectures can be compared or utilized in our NeuralGW, but one of our major contributions is the MMD layer. We have provided ablation studies to show the significance of this layer.
>
> [1] Zhang, W., Wang, Z., Fan, J., Wu, H., \& Zhang, Y. (2024). Fast Gradient Computation for Gromov-Wasserstein Distance. arXiv preprint arXiv:2404.08970.
>
> [2] Courty, N., Flamary, R., \& Ducoffe, M. (2017). Learning wasserstein embeddings. arXiv preprint arXiv:1710.07457.
>
> [3] Haviv, D., Kunes, R. Z., Dougherty, T., Burdziak, C., Nawy, T., Gilbert, A., \& Pe’Er, D. (2024). Wasserstein wormhole: Scalable optimal transport distance with transformers. ArXiv, arXiv-2404.
>
> __Response to Weakness 6__ (**writing**):
>
> We are sorry for the poorly written sentences. The corresponding sentences have been revised, and we have thoroughly polished the manuscript to enhance its clarity and presentation quality.

---

### Official Review · Reviewer_mnXv · 2025-10-29

**Soundness:** 2
**Presentation:** 2
**Contribution:** 2
**Rating:** 4
**Confidence:** 4

**Summary:**

This article proposes a neural network architecture for learning the Gromov–Wasserstein (GW) distance. The approach consists of embedding pairs of graphs using the concatenation of two GNNs—a GIN and a Graph Transformer—and then computing distances via an MMD over the embeddings with multiple kernel parameters, followed by a final MLP that predicts the GW distance. The authors describe the architecture in detail and validate their method on small graph datasets.

**Strengths:**

The paper addresses an interesting and important problem: scaling up the computation of the GW distance. While the use of neural networks to approximate Wasserstein distances has been explored in several prior works, extending this idea to the Gromov–Wasserstein setting is relatively novel. This formulation provides a natural way to handle graphs of varying sizes.
Some empirical results seem also to show taht the approach is working well against another NN approach for GW.

However, the paper currently feels somewhat incomplete, which prevents me from giving a clear recommendation for a clear acceptance.

**Weaknesses:**

- About the experiments:

The experimental section appears somewhat limited, and several aspects of the setup are unclear.

Overall, the experiments mostly evaluate how close the predicted distance is to the POT GW distance. While this is a reasonable starting point, it remains a limited and a bit narrow evaluation.
A more insightful analysis of the approximation error would strengthen the paper. For example, experiments on synthetic data, such as Stochastic Block Models (SBMs) with varying numbers of communities, could illustrate whether the learned distance preserves meaningful structure. A heatmap or distance matrix visualization could reveal clustering among graphs with the same community structure.
Similarly, scatter plots comparing “POT GW distance” versus “Neural GW distance” on held-out test pairs (not used during training) could demonstrate whether the predicted distances correlate well beyond the average error.

Regarding the spectral clustering experiment, I am not fully convinced by the results. Without simple baselines, it is difficult to assess whether the proposed method provides real benefits. How do standard spectral clustering methods with simple kernel on graphs perform on the same datasets?

On the clarity of the experiments: the meaning of “in-prediction” is not clearly defined. Is the error computed on pairs of graphs seen during training, or on unseen test pairs? This distinction is essential to assess generalization. The setup of sampled GW is also insufficiently explained. For instance, how is the number of sampled points chosen for the sampled GW baseline? Additionally, the authors should clarify the statement “...can be the adjacency matrices or the shortest path matrices”: which one is used in the reported experiments? It should also be emphasized that the so-called “true GW” distances are computed with POT — currently mentioned in a footnote. The most relevant comparison, in my view, would be against Wormhole, which shows that indeed the Neural GW has a interest.

- About the generalization error bound:

Although the theoretical result on the generalization bound is potentially interesting, its presentation is somewhat “dry”. The statement is not discussed or contextualized, and its concrete significance is unclear. The result appears to be a bound between an expectation and its empirical average — a type of inequality that is very standard in statistic and learning theory.
What makes this particular result specific to GW learning? why is this of particular interest? Is the proof technique unique to this setting? These points should be clarified. Moreover, the notations are particularly dense and difficult to parse, and there are several typos in the theorem statement.



- Minor remarks:

Table 4 is not very readable, as it is difficult to clearly observe how the error evolves with the number of layers. A line plot would be more appropriate, including the naive Neural GW baseline for reference.

The article also contains several awkward or imprecise formulations, for example:
    - Some sentences are a bit awkward or imprecise:
        - "These datasets vary in size from small to big.": Datasets with a maximum of 2000 nodes hardly qualify as "big."
        - "If the model designed does not match the data well, we will encounter significant overfitting or underfitting"
        - "While a graph represents the relationships among entities, the relationships and entities of graphs from different domains have different meanings and sizes, which results in huge challenges in cross-domain learning and generalization"

Finally, a recent relevant reference that could be cited (though not necessarily compared against, as it is very new) is [1].

[1] Unsupervised Learning for Optimal Transport plan prediction between unbalanced graphs, Sonia Mazelet, Rémi Flamary, Bertrand Thirion, NeurIPS, 2025.

**Questions:**

see above

---

> ### Author Response · Authors · 2025-11-28
>
> The authors sincerely thank you for recognizing our work. We provide the responses to your five questions as follows.
>
> __Response to Weakness 1__: About the experiments
>
> Thanks for your kind suggestion. We employed a Stochastic Block Model (SBM) to generate 200 graphs belonging to 5 classes, with 40 graphs per class. All graphs in 5 communities, while 5 classes differ in their underlying community structures. For illustration, we include five representative graphs (one from each class) to highlight the structural differences across classes (see Figure 3 in our revised paper).
>
> These 200 graphs were then split into training, validation, and test sets for the in-domain prediction task using NeuralGW. The full pairwise distance matrix, as well as the training, validation, test submatrices, are provided in Figure 4 in our revised paper. One can see that the distributions of distances in the training, validation, and test sets are generally consistent with that of the full distance matrix.
>
> * We train NeuralGW on the training set and plot the averaged predicted GW distance matrix on the test set. As shown in Figure 5 in our revised paper, the learned distances exhibit clear clustering patterns among graphs sharing the same community structure. This indicates that the learned GW distance successfully preserves meaningful structural information.
> * We also provide a scatter plot comparing POT GW distances with NeuralGW-predicted distances on held-out test pairs (rightmost plot in Figure 5 in our revised paper). The correlation coefficient is close to 0.95, demonstrating that the predicted distances align well with the ground truth beyond the average error.
>
> Thank you again for this valuable suggestion. We have incorporated these results into the paper.
>
> __Response to Weakness 2__: Regarding the spectral clustering experiment
>
> Thanks for pointing out this issue. We provide the results of standard spectral clustering with three more graph kernels, including the random walk kernel, Weisfeiler-Lehman kernel, and Weisfeiler-Lehman optimal assignment kernel in Table 5. Our method can still outperform them in most cases. We have added these new results to the revised PDF.
> |                              | ACC                        | NMI               | ARI                        | RI                         |
> |------------------------------|----------------------------|-------------------|----------------------------|----------------------------|
> | random walk kernel           | 0.7766                     | $\textbf{0.3082}$   | 0.3026                     | 0.6512                     |
> | WL kernel                    | 0.7074                     | 0.1360            | 0.1678                     | 0.5839                     |
> | WL-optimal assignment kernel | 0.6809                     | 0.1774            | 0.1237                     | 0.5631                     |
> | NeuralGW (PTC\_FM)           | $\textbf{0.8245}\pm0.0100$ | $0.2937\pm0.0264$ | $\textbf{0.4118}\pm0.0271$ | $\textbf{0.7092}\pm0.0129$ |
> | NeuralGW (BZR)               | $0.6777\pm0.2017$          | $0.2030\pm0.1307$ | $0.2464\pm0.1659$          | $0.6262\pm0.0798$          |
> | NeuralGW (PTC\_FM+BZR)       | $\textbf{0.8245}\pm0.0065$ | $0.2885\pm0.0163$ | $0.4101\pm0.0179$          | $0.7091\pm0.0086$          |
>
> __Response to Weakness 3__: On the clarity of the experiments
>
> Thanks for pointing out these issues. We are very sorry for the unclear description. We have revised the manuscript accordingly. Our clarifications are as follows.
>
> * __In-domain prediction vs. cross-domain prediction__: Both tasks are used to assess the effectiveness of the proposed NeuralGW. In-domain prediction refers to splitting a single dataset into training and test sets. Cross-domain prediction means that the model is trained on one or multiple datasets and evaluated on a different dataset.
> * __Error computation__: All reported errors are computed on unseen test pairs.
> * __Sampled GW (POT)}__: We adopted the default POT configuration for Sampled GW, where 100 points are sampled. This setting is appropriate for the datasets used in our experiments.
> * __Cost matrix__: We used adjacency matrices as the cost matrix of GW. Shortest path matrices are a reasonable alternative and represent an valuable direction for future investigation.
> * __POT implementation (GW)__: We clarified, in the main context rather than in a footnote, that the POT implementation is used for computing GW.
>
> We thank the reviewer again for the careful and constructive feedback. Details have been included in Appendix B.1 under the experimental settings.

---

> > ### Author Response · Authors · 2025-11-28
> >
> > __Response to Weakness 4__: About the generalization error bound
> >
> > Due to the space limitations in the initial submission, we didn't include a discussion for the theorem. The revised paper can include one more page; we include the discussion (highlighted in blue) at the end of Section 3.3. We use the theorem to show the impact of the GIN and transformer architectures, the position encoding, adjacency matrices, and the MMD layer on the generalization for GWD prediction. Although the proof used some standard techniques, such as Rademacher complexity, the entire configuration is specific to our model. For instance, the theorem shows that the number of MMDs has a tiny influence on the generalization, meaning that we can use a large $S$ to improve expressiveness, thereby reducing the prediction error. This is further verified by Figure 2.
> >
> > __Response to Weakness 4__: Minor remarks
> >
> > **Table 4 is not very readable**
> >
> > Thank you for the helpful suggestion. We have added a line plot for the error evolution. As shown in Figure 2, using multiple MMDs in the MMD layer consistently improves the performance of NeuralGW compared with using a single MMD or no MMD.
> >
> > **awkward or imprecise formulations**
> >
> > We are sorry for this. We have carefully polished the paper to enhance its clarity and quality.
> >
> > **New literature could be cited**
> >
> > Thank you for bringing this to our attention. We have added the corresponding citation in Section 4.2.

---

### Official Review · Reviewer_u6um · 2025-11-02

**Soundness:** 2
**Presentation:** 2
**Contribution:** 2
**Rating:** 4
**Confidence:** 5

**Summary:**

The authors propose to estimate the Gromov-Wasserstein (GW) distance between any pair of graphs with siamese graph neural network architectures, called NeuralGW-naive and NeuralGW, as well as an extension mostly discussed in the supplementary material to estimate transport plans. To this end, they propose to first encode each graph via a MPNN (GIN) and transformers operating on SVD-based positional encoding, providing node embeddings accounting for local and global structures respectively, before concatenating them to get final node embeddings. The latter are generated for two graphs independently, then node embeddings are compared across graphs, either using an Euclidean distance after a global (mean or sum) pooling step for NeuralGW-naive, or MMD distances with Gaussian kernels for NeuralGW. Authors provide a generalization bound for their approach and show that NeuralGW outperforms several baselines to estimate GW within a given domain and study the transferability of the models across different datasets.

**Strengths:**

-	Authors adapt siamese networks to the graph domain for the estimation of GW distance with an original pooling strategy to compare node embeddings of each graph leveraging MMDs.
-	They provide a theoretical analysis of the generalization of their models
-	Authors benchmark their approach against stochastic estimators for GW (Sampled GW, Spar-GW) and a deep learning baseline (Wormhole) and achieves the best GW estimations on several small-scale datasets.
-	They study the transferability of their models across different datasets showing rather good results.
-	They provide an ablation study over the number of MMD kernels considered in NeuralGW, showcasing that their approach is rather robust to this hyperparameter and outperforms other baselines in many settings.

**Weaknesses:**

- **W1. clarity of the paper.** Overall, I believe that the clarity of the introduction and certain sections should be improved on several aspects:
    -  In the context of OT including GW, graphs are always empirical distributions so it might be clearer to refer to methods which rely on the design of node embeddings in the second paragraph, either non-parametric ones like Weisfeilher-Lehman variants or deep learning based ones.
    -  Most theoretical results for GW hold for any network following [A] and its not clear in the context of the paper why measurable metric spaces are more relevant as methods operate on arbitrary adjacency matrices. This could be used to clarify the 3rd paragraph as well as Section 2.1.
    - In the 4th paragraph, the reference to FGW does not seem particularly appropriate as its goal is not to estimate GW. However, solvers to estimate exactly GW instead of an entropically regularized variant, such as the PPA from [B] or BAPG from [C] should be considered in the paper (in the introduction but also Section 4.1).
    - Many arguments are made w.r.t computational complexity in Section 2, so it could be nice to refer the reader to Section 3.4 at an earlier stage in the paper.
    - It can be clearer to formulate pseudo metric properties of neuralGW as a proposition in the main paper with proofs in the supplementary material.


- **W2. choice of architecture** : authors propose an architecture where local and global node features are learned by two independent encoders (MPNN and transformer), whereas many recent models propose instead to fuse both local and global information within each layer as well summarized in [D]. Authors should further justify their choices and potentially include such well established graph transformer architectures in their study.


- **W3. pooling mecanism** :
    - could authors clarify their choices for MMD compared to other kernels such as Wasserstein or Sinkhorn ? Could you also explain how did you select the parameters $\gamma_s$ for each kernel in the experiment section, for the in-domain and across-domain benchmarks as well as the ablation study on the number of kernels ?
    - It seems relevant to refer to template-based approaches which also mix GNN with kernels, for more informative global pooling than in neuralGW-naive as well as related works for the paper [E, F]

- **W4. potentially incomplete baselines**: I am concerned by the comprehensiveness of the literature review made by authors on estimators for GW in Section 4.2 and Section 5.
   - Courty & al (2017) focus on Wasserstein estimators hence i am not sure that the reference is properly used.
   - Zhang & al (2024) mentioned in the paper seems to be a highly performant competitor for GW estimation, it seems like it should be included in the benchmark.
   - Most neural estimators for GW discussed in [G] including GENOT from Klein & al (2024) seem relevant to address the task at end, could authors whether there were well-founded reasons to omit them from the benchmark ?


- **W5. relevance of the loss function**: Overall it is not clear to me whether picking an estimator of the GW distance as ground truth is a good global objective compared to methods which estimate directly the transport plan before plugging it within the GW loss. The former choice makes highly sensitive the model to the initial quality of the GW estimation, implying that the model is very likely learned with "label noise", which is not discussed in the paper. Hence could authors clarify how did they estimate the ground truth GW distances and clarify why they did not consider more robust learning strategies ?

- **W6. Transferability benchmark**: The choice of the different datasets used for learning models and evaluating them in Section 5.2 is not clear. It seems more natural to simply take models learned in the benchmark of Section 5.1 on each dataset individually and evaluate them on the other unseen datasets. Could authors provide a such benchmark ?  I guess models struggle to generalize while learning on very few datasets which motivated the choice of authors to learn from combination of datasets. However it seems that this study was only done considering datasets of molecules, which might have very similar structures. Wouldn't it be more relevant to stress diversity in the chosen datasets e.g a molecular one, a bio-informatic one and social network one to learn more transferable models ?

Tipos:
-  L35. By graph search, do you mean nearest graph search ? otherwise from my understanding graph search or traversal relates more to node-level tasks which might be a bit out of the scope of this paper.
-  L73. “are lack of” -> “lack of”
-  L181. It seems confusing to refer to the l1 loss here while I guess the l2 loss was used in experiments as it is used in the theoretical analysis.

[A] Chowdhury, S., & Mémoli, F. (2019). The Gromov–Wasserstein distance between networks and stable network invariants. Information and Inference: A Journal of the IMA, 8(4), 757-787.

[B] Xu, H., Luo, D., Zha, H., & Duke, L. C. (2019, May). Gromov-wasserstein learning for graph matching and node embedding. In International conference on machine learning (pp. 6932-6941). PMLR.

[C] Li, J., Tang, J., Kong, L., Liu, H., Li, J., So, A. M. C., & Blanchet, J. A Convergent Single-Loop Algorithm for Relaxation of Gromov-Wasserstein in Graph Data. In The Eleventh International Conference on Learning Representations.

[D] Rampášek, L., Galkin, M., Dwivedi, V. P., Luu, A. T., Wolf, G., & Beaini, D. (2022). Recipe for a general, powerful, scalable graph transformer. Advances in Neural Information Processing Systems, 35, 14501-14515.

[E] Chen, B., Bécigneul, G., Ganea, O. E., Barzilay, R., & Jaakkola, T. (2020). Optimal transport graph neural networks. arXiv preprint arXiv:2006.04804.

[F] Vincent-Cuaz, C., Flamary, R., Corneli, M., Vayer, T., & Courty, N. (2022). Template based graph neural network with optimal transport distances. Advances in Neural Information Processing Systems, 35, 11800-11814.

[G] Carrasco, X. A., Nekrashevich, M., Mokrov, P., Burnaev, E., & Korotin, A. (2023). Uncovering Challenges of Solving the Continuous Gromov-Wasserstein Problem. arXiv preprint arXiv:2303.05978.

**Questions:**

I invite authors to discuss the weaknesses mentioned above and have a last question to clarify computational performances of their neuralGW model:
- Could you detail which device (CPU, GPU etc) was used in the benchmark in supplementary B.2 ?

---

> ### Author Response · Authors · 2025-11-28
>
> The authors sincerely appreciate the insightful comments made by the reviewer.
>
> __Response to W1__:
>
> * In the updated PDF, we have revised the second paragraph (highlighted in blue) according to your suggestion.
> * We agree that most theoretical results for GW extend to arbitrary networks, e.g. weighted and directed graphs, as shown in [A]. In our work, we focus on undirected graphs, for which the adjacency matrix naturally induces a metric structure that aligns well with the standard GW formulation on measurable metric spaces. Although in our experiments, the graphs we considered are unweighted, our NeuralGW can be applied to weighted graphs directly. Since our methods operate directly on these adjacency-based structures, the measurable metric space viewpoint provides a convenient and sufficiently general framework for our setting. To make this clearer, we have revised the 3rd paragraph and updated Section 2.1 to better articulate this connection.
> * Thanks for your valuable suggestion. We have revised the 4th paragraph and Section 4.1. PPA [B] and BAPG [C] are both discussed in those parts.
> * We have added such a cross-reference in the 5th paragraph.
> * Sure. We have formulated it as Proposition 1 and included the proof in Appendix A.

---

> ### Author Response · Authors · 2025-11-28
>
> __Response to W2__:
>
> Thanks for your valuable suggestion. [D] proposed a modular framework, GraphGPS, which integrates both local and global structural information within each layer by combining message-passing (MPNN) with a global attention mechanism. Given its expressive design, the GraphGPS layer can serve as an alternative to our parallel GIN-Transformer architecture. To evaluate this possibility, we replaced our GIN-Transformer module with a GraphGPS layer and introduced two corresponding variants:
>
> * $\text{NeuralGW}_\text{GPS}$, the GraphGPS-based counterpart of NeuralGW, and
> * $\text{NeuralGW}_\text{GPS}$-Naive, the counterpart of NeuralGW-Naive where the MMD layer and final MLP are removed.
>
> We compared these variants alongside our original methods and other baselines under both in-domain and cross-domain predictions. The numerical results, reported in Tables 1 and 2, show that our proposed architectures consistently outperformed their GraphGPS-based counterparts in most cases. One possible reason is that, in GraphGPS, the within-layer fusion of pure local and pure global structural information may lead to difficulty in extracting local and global information from graphs. As far as we are concerned, local information and global information are very different and should be extracted separately, and finally be stacked to predict the GW distance.
>
> __Table 1__: Relative error of in-domain prediction. The best result is marked in bold.
> |                                    | MUTAG                      | BZR                        | COX2                       | DHFR                       |
> |------------------------------------|----------------------------|----------------------------|----------------------------|----------------------------|
> | Sampled GW                         | 0.4527                     | 0.2916                     | 0.4256                     | 0.3223                     |
> | Spar-GW (8n)                       | 0.7677                     | 1.0872                     | 1.3896                     | 1.0398                     |
> | Spar-GW (4n)                       | 0.6991                     | 1.1227                     | 1.5138                     | 1.1161                     |
> | Spar-GW (2n)                       | 0.7982                     | 1.1436                     | 1.4421                     | 1.1104                     |
> | Wormhole ($\beta=1$)               | $0.5416\pm0.0194$          | $0.5029\pm0.0156$          | $0.5908\pm0.0304$          | $0.4909\pm0.0098$          |
> | Wormhole ($\beta=0$)               | $0.5046\pm0.0171$          | $0.4142\pm0.0216$          | $0.3997\pm0.0098$          | $0.4238\pm0.0058$          |
> | $\mathrm{NeuralGW}_\mathrm{GPS}$-Naive | $0.4943\pm0.0741$          | $0.4277\pm0.0306$          | $0.5189\pm0.1009$          | $0.4603\pm0.0674$          |
> | $\mathrm{NeuralGW}_\mathrm{GPS}$       | $0.2427\pm0.0170$          | $0.2104\pm0.0024$          | $\textbf{0.1410}\pm0.0025$ | $0.2033\pm0.0182$          |
> | NeuralGW-Naive                     | $0.3087\pm0.0106$          | $0.3036\pm0.0157$          | $0.3108\pm0.0179$          | $0.2575\pm0.0075$          |
> | NeuralGW                           | $\textbf{0.1414}\pm0.0164$ | $\textbf{0.1779}\pm0.0445$ | $0.1466\pm0.0075$          | $\textbf{0.1267}\pm0.0077$ |
>
> __Table 2__: Relative error of cross-domain prediction
> | Trainsets                          | $\mathrm{BZR + PTC\\_FM}$     | $\mathrm{BZR + PTC\\_FM}$ | $\mathrm{BZR + COX2}$        |
> |------------------------------------|----------------------------|------------------------|----------------------------|
> | Testset                            | MUTAG                      | DHFR                   | DHFR                       |
> | Sampled GW                         | 0.4527                     | $\textbf{0.3223}$        | 0.3223                     |
> | Spar-GW (8n)                       | 0.7677                     | 1.0398                 | 1.0398                     |
> | Spar-GW (4n)                       | 0.6991                     | 1.1161                 | 1.1161                     |
> | Spar-GW (2n)                       | 0.7982                     | 1.1104                 | 1.1104                     |
> | Wormhole ($\beta=1$)               | $0.4141\pm0.0063$          | $0.6737\pm0.0449$      | $0.7157\pm0.0649$          |
> | Wormhole ($\beta=0$)               | $0.3978\pm0.0109$          | $0.5524\pm0.0314$      | $0.5894\pm0.0406$          |
> | $\mathrm{NeuralGW}_\mathrm{GPS}$-Naive | $0.5112\pm0.1376$          | $0.5176\pm0.1566$      | $0.3877\pm0.0137$          |
> | $\mathrm{NeuralGW}_\mathrm{GPS}$       | $0.3116\pm0.0416$          | $0.4321\pm0.1620$      | $0.2764\pm0.0186$          |
> | NeuralGW-Naive                     | $0.3806\pm0.0321$          | $0.3912\pm0.0246$      | $0.3583\pm0.0319$          |
> | NeuralGW                           | $\textbf{0.2280}\pm0.0158$ | $0.3439\pm0.0919$      | $\textbf{0.1521}\pm0.0479$ |

---

> ### Author Response · Authors · 2025-11-28
>
> __Response to W3__:
>
> **Question on the MMD layer**
>
> We choose to use MMD for NeuralGW primarily because it is computationally more efficient than Wasserstein or Sinkhorn, especially when applied repeatedly during neural network training. This efficiency makes MMD a more practical choice for our framework.
>
> Regarding the selection of kernel parameters $\gamma_1,\gamma_2,\ldots,\gamma_S$, we let $\gamma_s=\gamma s^2$, $s=1,2,\ldots,S$, where  $\gamma$ is a learnable scale parameter initialized at 0.001. That means, for the 10 MMDs, we let the kernel parameters be $[\gamma\times 1^2,\gamma\times 2^2,\ldots,\gamma\times 10^2]$, respectively. This setting has been used in all experiments.
>
> **Question on template-based approaches for [E,F]**
>
> We summarize the difference between our NeuralGW and OT-GNN [E]/TFGW-GNN [F] as follows.
> * From the perspective of the task, both OT-GNN [E] and TFGW-GNN [F] are developed for the graph classification task, while our NeuralGW aims at approximating the GW distance.
> * In terms of the network architecture, our NeuralGW is indeed a Siamese neural network, which is different from OT-GNN and TFGW-GNN.
> * Taking input information into consideration, both OT-GNN and TFGW-GNN have to take the node features as part of their input, while only the adjacency matrix is required in NeuralGW.
>
> Thus, our method has a significant difference from [E,F]. We have added the two references in Introduction.
>
> __Response to W4__:
>
> **Question on the work in Courty et al. (2017)**
>
> Thanks for raising the concern.
> Indeed, Deep Wasserstein Embedding (DWE) in Courty et al (2017) can approximate the GW distance. However, it is a CNN-based embedding algorithm for gray-scale images. This limits its applicability because it can only embed 2D/3D images or point clouds with fixed-grid support. In other words, DWE cannot embed cohorts of point clouds in arbitrary space and high dimensions that cannot be voxelized. Due to this, DWE cannot be a proper baseline under the scope of our topic.
>
> **Question on the work in Zhang et al. (2024)**
>
> Zhang et al (2024) focus on the acceleration of **entropic** GW by proposing a fast gradient computation method, which is incompatible with our topic to some extent. In this sense, we did not include it in the benchmark. The distance given by entropic GW is much larger than GWD if the entropic regularizer is not small enough. Actually, we have some results of the entropic GW, but we haven't included them in our initial submission. Here, we show some results in the following table, where the entropic regularizer is set to 0.3, 0.1, or 0.01. We see that the error is much higher than that of our NeuralGW. Using a smaller regularization hyperparameter may reduce the error but often leads to unstable or very slow optimization. By the way, it is quite difficult to determine the best regularization hyperparameter for the trade-off between accuracy and efficiency.
>
> __Table 3__: Average relative error of in-domain prediction
> |                                      | MUTAG             | BZR               | COX2              | DHFR              |
> |--------------------------------------|-------------------|-------------------|-------------------|-------------------|
> | $\text{Entropic GW}_{\epsilon=0.3}$  | 1.7234            | 1.7383            | 2.2125            | 1.7303            |
> | $\text{Entropic GW}_{\epsilon=0.1}$  | 1.1008            | 1.5939            | 1.9806            | 1.5739            |
> | $\text{Entropic GW}_{\epsilon=0.01}$ | 0.2124            | 1.0613            | 1.9609            | 1.4273            |
> | NeuralGW                             | $0.1414\pm0.0164$ | $0.1779\pm0.0445$ | $0.1466\pm0.0075$ | $0.1267\pm0.0077$ |
>
>
> We have included the two references you mentioned in our paper.
>
> **Question on other estimators for GW in [G] including GENOT**
>
> Our work focuses on the neural approximation of discrete GW distances between graphs. Five methods in [G] are discussed as representatives of solving continuous GW. Four of them, including StructuredGW (Sebbouh et al., 2024), FlowGW (Klein et al., 2023), AlignGW (Alvarez-Melis \& Jaakkola, 2018), and RegGW (Uscidda et al., 2024), heavily rely on Sinkhorn acceleration in entropic GW, whereas CycleGW (Zhang et al., 2021) proposed the unbalanced bidirectional Gromov-Monge divergence. All of them have a significant difference from our NeuralGW. Therefore, we excluded them from our benchmark. We have included these references and a discussion in Section 4.2 of our revised paper.

---

> ### Author Response · Authors · 2025-11-28
>
> __Response to W5__:
>
> Thank you very much for this insightful comment. In general, it is intuitively more difficult to directly estimate the transport plan ($n^2$ high-dimensional label) before plugging it into the GW loss than picking an estimator of the GW distance (a single scalar label) as the ground truth. Furthermore, since GW yields a constrained non-convex quadratic minimization problem, it is hardly possible to attain the global minimum and get the optimal transport plan. Although the POT package we used does not guarantee optimal solutions, it is a standard Python library commonly used in the literature and can provide reliable results.  The label could be noisy, but we expect that the variance is not significant.  We therefore take the results from POT as the ground truth GW distances for model training and the subsequent comparison analysis.
>
> __Response to W6__:
>
> Thanks for your thoughtful suggestion. In the following table, we provide the numerical results of the cross-domain prediction, where one dataset (rows) is for training and three datasets (columns) are for testing. We observe that the NeuralGW trained on other datasets performs better than the NeuralGW trained on MUTAG. One reason is that MUTAG (with 188 graphs) is much smaller than the other three datasets (each with more than 400 graphs).
>
> __Table 4__: Cross-domain prediction from a single dataset (row/train) to another single dataset (column/test)
> |                  | MUTAG             | BZR               | COX2              | DHFR              | AVG    |
> |------------------|-------------------|-------------------|-------------------|-------------------|--------|
> | trained on MUTAG | $0.1414\pm0.0164$ | $0.4569\pm0.0679$ | $0.9063\pm0.2501$ | $0.6575\pm0.0379$ | 0.5405 |
> | trained on BZR   | $0.3658\pm0.0475$ | $0.1779\pm0.0445$ | $0.1970\pm0.0360$ | $0.1832\pm0.0494$ | 0.2305 |
> | trained on COX2  | $0.5691\pm0.0434$ | $0.3147\pm0.0111$ | $0.1466\pm0.0075$ | $0.2497\pm0.0280$ | 0.3200 |
> | trained on DHFR  | $0.5333\pm0.0251$ | $0.1866\pm0.0168$ | $0.1426\pm0.0027$ | $0.1267\pm0.0077$ | 0.2473 |
>
> To evaluate the impact of diverse training data, we constructed a new set comprising molecular, bioinformatics, and social network datasets, as suggested. Comparative results against a model trained solely on molecular data are presented in the table below. The integrated dataset enhanced predictive performance in two out of three cases, demonstrating the utility of incorporating related domain data.
>
> Since the computation of the ground-truth GW distances is time-consuming, we currently have a small piece of results. We are going to include more results in the revised paper.
>
> __Table 5__: Cross-domain prediction from multiple datasets (row/train) to a single dataset (column/test)
> |                                       | DHFR              | ENZYMES  | IMDB              |
> |---------------------------------------|-------------------|-------------------|-------------------|
> | MUTAG                                 | $0.6575\pm0.0379$ | $0.5985\pm0.0918$ | $0.8015\pm0.1703$ |
> | MUTAG + DD + COLLAB | $0.6877\pm0.1247$ | $0.2301\pm0.1743$ | $0.3428\pm0.1672$ |
>
> __Response to Typos__:
>
> Thank you for pointing this out. We have corrected them and further polished the manuscript accordingly.
>
> __Response to Questions__:
>
> We perform all methods on CPU to ensure a fair comparison.

---

### Author Response · Authors · 2025-12-03

Dear AC,

We sincerely thank you and all reviewers for your time and constructive feedback. We have carefully responded to all comments given by the reviewers and revised the paper accordingly.

Although the ratings are 4, all reviewers recognized the novelty and contribution of our method. For instance:

* Reviewer u6um highlighted the core strengths of our work, e.g., "__showing rather good results__" and "__outperforms other baselines in many settings__".
* Reviewer mnXv noted that "__the paper addresses an interesting and important problem: scaling up the computation of the GW distance.__"
* Reviewer qoyc emphasized that "__the paper addresses an important topic, which is the estimation of graph distances.__"
* Reviewer ioen acknowledged that "__without no doubt, the line of research proposed by the authors is of interest, since a direct computation of the optimal transports distances, in general, is quite prohibitive.__"


The Questions and concerns raised by the reviewers primarily relate to clarifying theoretical details and providing additional supporting experiments.

Below, we highlight the key updates made for each reviewer.

1. Reviewer u6um raises concerns about the paper's clarity and the justification of key design choices, including the architecture, pooling method, baselines, and transferability evaluation.
* We formalized the pseudo-metric properties of NeuralGW as Proposition 1 and added the proof in the supplementary material;
* We added a comparison experiment between NeuralGW and $\text{NeuralGW}_\text{GPS}$;
* We clarified our rationale for using MMD as the readout layer;
* We expanded the transferability benchmark, including (i) cross-domain predictions from a single dataset to another single dataset, and (ii) predictions from a combined set of molecular, bio-informatic, and social network datasets to an unseen dataset.

2. Reviewer mnXv argues that the experiments are limited and insufficient especially on synthetic data, lack simple but basic baselines, and that the theoretical result is poorly contextualized and difficult to interpret.
* We added visualizations of NeuralGW on random graphs generated by the Stochastic Block Model (SBM);
* We incorporated additional baselines: standard spectral clustering with the random walk kernel, Weisfeiler-Lehman (WL) kernel, and WL-optimal assignment kernel;
* We elaborated on the implications of the generalization bound specific to GW learning and added a line plot illustrating error evolution as the number of MMDs in the MMD layer increases.

3. Reviewer qoyc highlights concerns about limited discriminative power, potential over-smoothing, unclear complexity assumptions, and insufficient baselines.
* We clarified the effectiveness of NeuralGW in graph learning tasks;
* We added more experimental results including an ablation study on the GIN-Transformer submodules and an analysis of the impact of the number of GIN layers, demonstrating that the NeuralGW architecture effectively mitigates over-smoothing;
* We provided more detailed explanations of the time complexity;
* We incorporated additional related literature into the paper.

4. Reviewer ioen finds minor issues in the pseudo-metric claim, the derivation of the generalization bound and the complexity analysis.
* We showed that NeuralGW is permutation invariant, which supports the reformulation of the pseudo-metric property as Proposition 1;
* We expanded the theoretical details for intermediate steps used in the generalization bound;
* We added additional explanations regarding the time complexity.

From the whole picture, we summarize the major revisions across the writing, theory, and experiments:

1. Writing and organization:

* The Introduction was improved by reorganizing the review of existing methods of pairwise graph comparison;
* We added further relevant literature in both the Introduction and Related Work sections, including Proximal Point Algorithm (PPA)-based method, Bregman Alternated Projected Gradient (BAPG)-based method, GraphGPS, OT-GNN, TFGW-GNN, Deep Wasserstein Embedding (DWE), FGC-GW, and other methods for solving continuous GW like StructuredGW, FlowGW, AlignGW, and RegGW;
* We provided more detailed descriptions of the experimental settings, including dataset splits, hyperparameter choices, and evaluation metrics;
* We corrected typographical errors throughout the paper.

2. Theoretical part:

* We established permutation invariance for NeuralGW and reformulated the pseudo-metric property as Proposition 1 with proof included in the supplementary material;
* We elaborated on the interpretation of the generalization bound specific to GW learning and added a line plot showing error evolution with respect to the number of MMDs;
* Additional details of the generalization bound proof were provided;
* We added clarifications about the time complexity.

---

> ### Author Response · Authors · 2025-12-03
>
> 3. Experimental part:
>
> * We added the comparison experiment of NeuralGW with $\text{NeuralGW}_\text{GPS}$;
> * We expanded the transferability experiments, including both single-dataset cross-domain prediction and multi-domain combinations;
> * We added visualizations of NeuralGW on SBM-generated graphs;
> * We included additional spectral clustering baselines with three classical graph kernels;
> * We added results from the ablation study on GIN-Transformer submodules and the influence of the number of GIN layers, showing that NeuralGW mitigates over-smoothing.
>
>
> We thank the AC and all reviewers once again for their thoughtful and constructive reviews.
>
> Best,
>
> Authors

---

### Meta-Review · Area_Chair_tLya · 2026-01-06

**Summary:**

In this submission, the authors proposed a new member of neural GW distance for graph comparison. The proposed method learns a neural network with an "MPNN+Transformer" architecture to encode graphs across different domains. Each graph is finally represented as an embedding vector, and accordingly, the GW distance between different graphs is approximated by the Euclidean distance between their embedding vectors. Experiments show the feasibility of the proposed method to some extent.

The main concerns of reviewers include: 1) the rationality of the model design (e.g., the MMD-based pooling), 2) the lack of solid analytic experiments, 3) the writing quality, and 4) the rationality of the experimental settings (e.g., the selection of baselines and datasets). Although the authors made efforts to revise the paper in the rebuttal phase, some concerns remain unresolved. In addition, I am curious whether a model trained on a single graph dataset can generalize to multiple graph datasets (e.g., computing the GW distance between graphs across different domains), which is necessary to verify the main claim of the authors. Therefore, I think this work requires one more round of review.

**Reviewer Concerns:**

The authors added more explanations and experiments in the revised paper. However, the rationality of the MMD-based pooling and the generalizability problem are not fully resolved at the current stage. In addition, I think the writing of the paper can be further improved.

**Reviewer Scores:**

I think the reviewers would have maintained their scores.

---

### Decision · Program_Chairs · 2026-01-26

Reject